# Biodiversity time series are biased towards increasing species richness in changing environments

**Lucie Kuczynski** [1] ✉, **Vicente J. Ontiveros** [2,3] **& Helmut Hillebrand** [1,4,5]

The discrepancy between global loss and local constant species richness has led to debates over data quality, systematic biases in monitoring programmes and the adequacy of species richness to capture changes in biodiversity. We show that, more fundamentally, null expectations of stable richness can be wrong, despite independent yet equal colonization and extinction. We analysed fish and bird time series and found an overall richness increase. This increase reflects a systematic bias towards an earlier detection of colonizations than extinctions. To understand how much this bias influences richness trends, we simulated time series using a neutral model controlling for equilibrium richness and temporal autocorrelation (that is, no trend expected). These simulated time series showed significant changes in richness, highlighting the effect of temporal autocorrelation on the expected baseline for species richness changes. The finite nature of time series, the long persistence of declining populations and the potential strong dispersal limitation probably lead to richness changes when changing conditions promote compositional turnover. Temporal analyses of richness should incorporate this bias by considering appropriate neutral baselines for richness changes. Absence of richness trends over time, as previously reported, can actually reflect a negative deviation from the positive biodiversity trend expected by default.

The expectation that species richness remains constant in the absence of external forcing at ecological time scales is deeply rooted in ecological theories[1,2] assuming a dynamic equilibrium between colonizations and extinctions[3]. Assessments of time series in the global change context thus interpret deviations from balanced dynamics such as positive and negative trends in species number as a response to improving or deteriorating environmental conditions, respectively[4,5]. Under increased environmental suitability (Fig. 1), most species will profit, and the expected positive trends emerge, although colonizations may also be delayed ('immigration credit'[6]). On the other hand, one can expect

that a reduction in habitat suitability will affect most species negatively up to the extinctions of some (Fig. 1). As the exponential decline of existing populations takes time (for example, because of plasticity, use of microrefugia), extinction debts will lead to a delayed reduction in richness[6,7] and the negative richness trends will only emerge later.

The scale- and effort-dependency of species richness as a metric creates uncertainty around trends[8,9], while, in addition, richness does not capture compositional turnover but rather the net difference between colonizations and extinctions[10,11]. Even more fundamentally though, the temporal response of richness might not match our

[1]Plankton Ecology Lab, Institute for Chemistry and Biology of the Marine Environment, Carl von Ossietzky University Oldenburg, Wilhelmshaven, Germany. [2]Institute of Aquatic Ecology, University of Girona, Girona, Spain. [3]Department of Life Sciences, Ben-Gurion University of the Negev, Be'er Sheva, Israel. [4]Helmholtz Institute for Functional Marine Biodiversity at the University of Oldenburg (HIFMB), Oldenburg, Germany. [5]Alfred-Wegener-Institute, Helmholtz Centre for Polar and Marine Research, Bremerhaven, Germany. ✉e-mail: lucie.kuczynski@hotmail.com

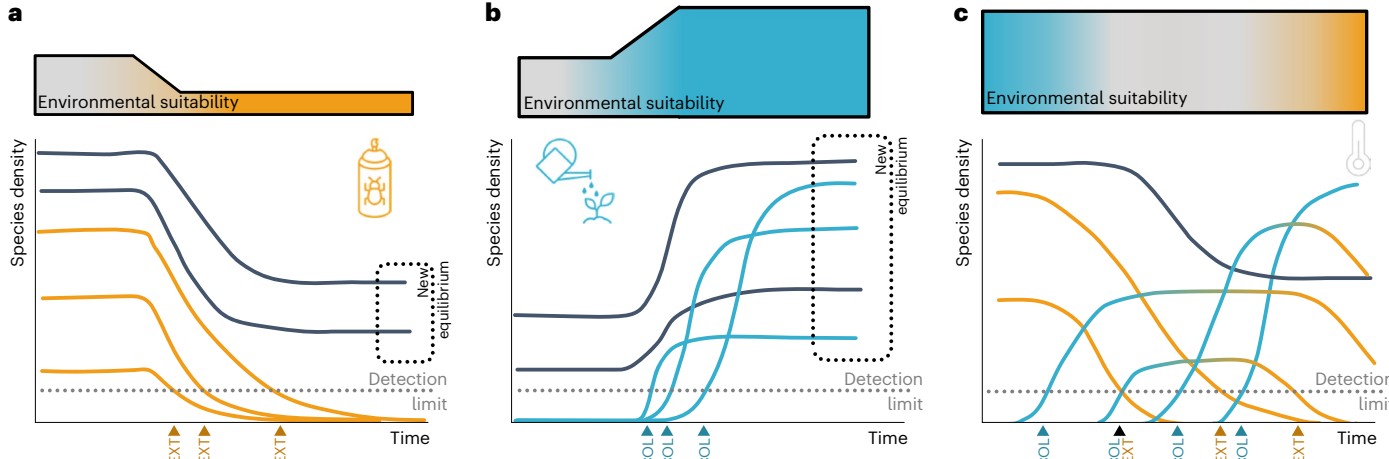

**Fig. 1 | Conceptual figure of the impact of different anthropogenic changes on species diversity and species density.** Yellow lines indicate species experiencing population declines up to extinction while blue ones indicate species experiencing increases in density. **a**, The case of a negative impact (for example, increase in pesticides, habitat fragmentation) resulting in a lower equilibrium richness, which can take some time to establish as declining populations persist (extinction debt). **b**, A clear positive impact (for example, enlargement of habitat size through restauration) that leads to a higher equilibrium richness, which might take time to establish as gained populations need some time to colonize (immigration credit). **c**, A steady change: even though as many species decline (that is, 'losers') as colonize (that is, 'winners'), the observed richness increases if new species arrive earlier than species go extinct. This increase does not disappear, as any new time segment added leads again to earlier colonizations than extinctions, with no new equilibrium being reached. EXT, extinction; COL, colonization.

expectation, especially if the environment-driven trajectory is not clearly negative or positive (Fig. 1a,b) but neutral, as some species are favoured and can colonize while others decline and eventually go extinct (Fig. 1c).

To conceptualize the issue, consider a neutral environmental change such that there are equal numbers of 'winners' and 'losers', and richness is expected to remain constant. However, under low dispersal limitation, one can assume that colonizations (defined as the first colonization event over a given time series) will be fast (as it needs only few propagules), whereas extinctions (defined as the last extinction event over a given time series) will be delayed because in the absence of catastrophic mortality population growth will slowly turn negative for the losers. For dominant species, the resulting decline in abundance will result in extinction after many generations. This extinction process might be further slowed down if density-dependent mortality declines or populations adapt their phenotypes to the new conditions. This bias towards earlier colonizations will result in increasing richness over time, which may be transient if the environmental change stops at some point such that colonizations and extinctions can equilibrate again. However, if environmental change continues, each incremental increase in observation time will allow further colonizations, resulting in further imbalance detected as increasing richness in finite time series (Fig. 1c). On the other hand, if a community exhibits a strong inertia in its dynamics, rare species are likely to go extinct and not locally recolonize. Thus, as the majority of species are rare, decrease in richness will emerge.

## Results and discussion
### Temporal trends in species richness
Here, we combine observational data and simulations to test whether this imbalance is strong enough to fundamentally shift species richness trends to slopes different from zero by default. We first analysed species richness trends using 3,036 European empirical freshwater fish community time series from the highly curated RivFishTIME dataset[12] (average duration = 24 years), along with 4,317 time series from the Breeding Bird Survey in North America[13] (average duration = 37 years; Methods). Across the empirically sampled communities, the average

slopes from the linear mixed-effects (LME) model were +0.02 (standard error (s.e.) = 0.001, $P < 0.001$, marginal $R^2 = 0.002$, conditional $R^2 = 0.85$) and +0.03 (s.e. = 0.0001, $P < 0.001$, marginal $R^2 = 0.007$, conditional $R^2 = 0.83$) for freshwater fish and breeding bird communities, respectively (Fig. 2a,d). The empirical data thus correspond to previous meta-analyses[4,10,14,15], showing no overall decline in local richness, but rather a small yet significant average increase over time.

Shorter time series revealed more variable estimates for slopes and larger standard errors (Fig. 3, Supplementary Figs. 3 and 4, and Supplementary Tables 4 and 5). To test whether the positive overall richness trend was driven by short time series only, we used a generalized additive model for location scale and shape (GAMLSS[16]; Methods). While only the variance in species richness trends was affected by time series length for freshwater fish (estimate$_{slope}$ ± s.e. = $1 \times 10^{-5} \pm 1 \times 10^{-5}$, $P = 0.3$; estimate$_{variance}$ ± s.e. = $-0.04 \pm 2 \times 10^{-3}$, $P < 0.001$; $R^2 = 0.20$), both the mean and the variance in species richness trends were impacted for birds (estimate$_{slope}$ ± s.e. = $1 \times 10^{-5} \pm 1 \times 10^{-6}$, $P < 0.001$; estimate$_{variance}$ ± s.e. = $-0.03 \pm 8 \times 10^{-4}$, $P < 0.001$; $R^2 = 0.29$; Fig. 3a,d). Thus, when dispersal is not strongly constraining communities (for example, avian communities), short time series exhibit a duration-related underestimation bias in the observed trends. While we fully acknowledge the time and money already needed to collect such data[17], we need to accept that most currently used worldwide long-term datasets actually capture relatively short time series[18,19]. Therefore, our results strongly suggest that short time series potentially underestimate diversity loss, as previously claimed[20].

We compared these observations with a null model for which we fully randomized the observed yearly chronosequences of species, thereby fully removing temporal autocorrelation from year to year in species dynamics. Such null models are often used to provide a benchmark for a given diversity metric in the absence of driving processes[21]. For both taxa-specific null models, species richness was steady over time (LME, fish: estimate ± s.e. = $-8 \times 10^{-5} \pm 3 \times 10^{-4}$, $P = 0.8$, marginal $R^2 < 0.001$, conditional $R^2 = 0.84$; birds: estimate ± s.e. = $-2 \times 10^{-4} \pm 1 \times 10^{-4}$, $P = 0.2$, marginal $R^2 < 0.001$, conditional $R^2 = 0.82$; Fig. 2b,e), while the variance was reduced under long time series (fish: estimate$_{variance}$ ± s.e. = $-5 \times 10^{-2} \pm 5 \times 10^{-4}$, $P < 0.001$, $R^2 = 0.30$; birds:

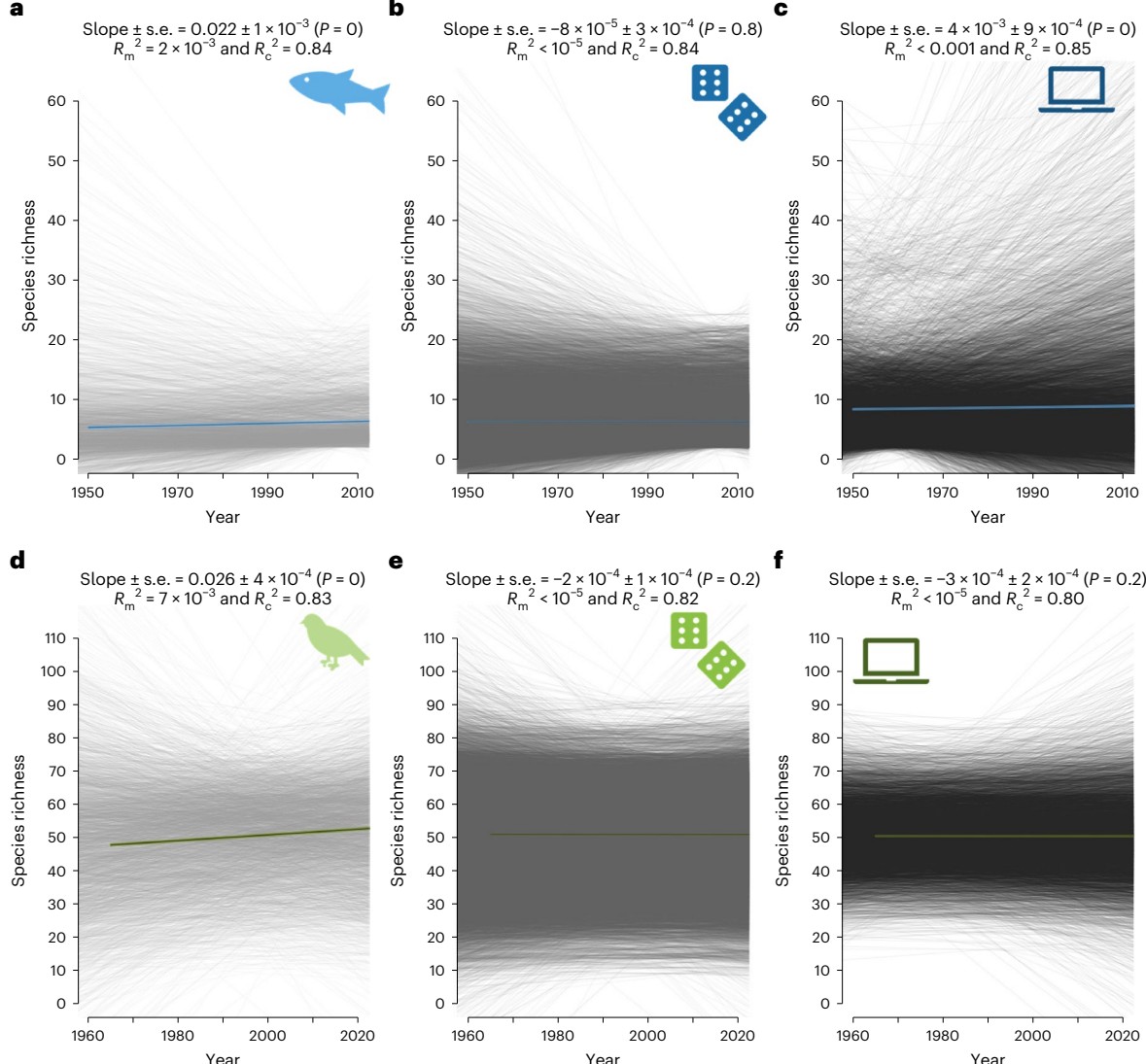

**Fig. 2 | Species richness over time for freshwater fish and breeding birds.**
**a–f**, Species richness over time for freshwater fish (**a–c**) and breeding birds
(**d–f**). Background lines are the empirical (**a,d**), null model based (**b,e**) and
simulated (**c,f**) trends in species richness estimated with a linear regression for

each individual site. Coloured lines are the output of the LME models (estimate ±
s.e.) from which estimates and goodness-of-fit are indicated on each panel. $R_m^2$,
marginal $R^2$; $R_c^2$, conditional $R^2$.

estimate$_{variance}$ ± s.e. = $-4 \times 10^{-2} \pm 3\,10^{-4}$, $P < 0.001$, $R^2 = 0.48$; Fig. 3b,e).
However, this classic null model approach is highly unrealistic for bio-
diversity time series as it allows any species to flip between absence,
rare and abundant occurrences, which does not occur in actual popula-
tions. In the absence of catastrophic extinctions, the population size
at any time point is correlated with the abundance at the previous time
step via the specific birth and death rates, resulting in strong temporal
autocorrelation under regular monitoring when sampling intervals are
not very large compared with generation time.

To analyse whether incorporating temporal autocorrelation mat-
ters for null expectations, we simulated 9,999 time series of neutral
communities. These simulations matched the empirical observations
with respect to mean and variance of time series length and species
richness. We derived these time series from a neutral model[22,23] based
on the theory of island biogeography[2], simulating species occurrences
while controlling for equilibrium richness and temporal autocorrela-
tion. We explored a large range of autocorrelations (Supplementary
Table 1), but highlight a case with an autocorrelation level match-
ing the observed temporal autocorrelation. Despite being a neutral

model, simulated time series for river fish exhibited increased spe-
cies richness over time (estimate ± s.e. = $4 \times 10^{-3} \pm 9 \times 10^{-4}$, $P < 0.001$,
marginal $R^2 < 0.001$, conditional $R^2 = 0.85$), which suggests that these
fish communities are not at equilibrium with their historical context
(Fig. 2c,f, Supplementary Figs. 1 and 2, and Supplementary Tables 2
and 3). By contrast, simulated time series for breeding birds did not
show a significant deviance from neutral trends (estimate ± s.e. = $-3
\times 10^{-4} \pm 2 \times 10^{-4}$, $P = 0.2$, marginal $R^2 < 0.001$, conditional $R^2 = 0.80$),
which may reflect that bird communities are less constrained in their
dispersal, allowing stronger rescue effect[24]. The simulated slope of
richness over time was significantly independent from time series
length (fish: estimate$_{slope}$ ± s.e. = $-6 \times 10^{-6} \pm 1 \times 10^{-5}$, $P = 0.6$, $R^2 = 0.20$;
birds: estimate$_{slope}$ ± s.e. = $2 \times 10^{-7} \pm 4 \times 10^{-7}$, $P = 0.7$, $R^2 = 0.48$), only
variance in species richness trends decreased with longer time series
(fish: estimate$_{variance}$ ± s.e. = $-5 \times 10^{-2} \pm 9 \times 10^{-4}$, $P < 0.001$; birds: estimat-
e$_{variance}$ ± s.e. = $-4 \times 10^{-2} \pm 5 \times 10^{-4}$, $P < 0.001$; Fig. 3c,f). This pattern holds
for most of the settings of autocorrelation and balance between colo-
nization and extinction we have tested (Supplementary Figs. 3 and 4,
and Supplementary Tables 4 and 5). Thus, the observed departure from

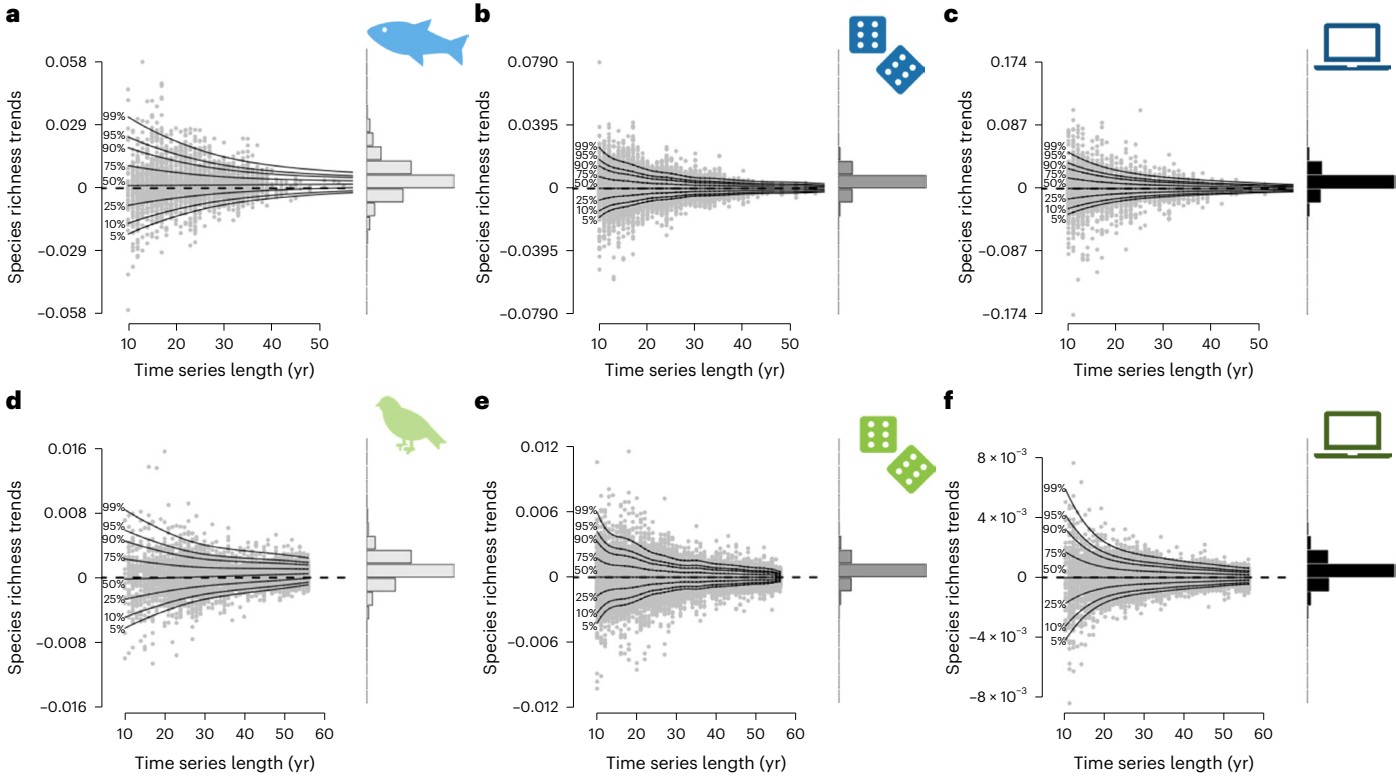

**Fig. 3 | Effect of time series length on species richness trends for observed, randomized and simulated data for riverine fish and breeding birds.**
**a**–**f**, Effect of time series length on species richness trends for observed (**a**,**d**), randomized (**b**,**e**) and simulated (**c**,**f**) data for riverine fish (**a**–**c**) and breeding birds (**d**–**f**). For each panel, on the right side, the distribution of the species richness trends is represented, and the solid black lines represent percentile curves estimated with GAMLSS.

a zero slope for simulated data, especially in the case of riverine fish, is not linked to the empirical time series being too short[25].

### Net imbalance between colonizations and extinctions

Richness increases are inevitable when population dynamics exhibit strong autocorrelation (for example, strong dispersal limitation), which may mask the true richness trends (expected to be null in the simulations), even for time periods substantially longer than our observations. Our simulations turn the interpretation of the RivFishTIME data around: on average, richness increases, but less than expected from a neutral community with similar autocorrelation. The observed positive trend thus is a negative deviation from the neutral expectation, meaning that colonizations happen slower and/or extinctions faster than needed to balance winners and losers. To test whether the bias towards positive richness trends is based on the imbalance between colonization and extinctions, we compared the cumulative number of colonizations ($C_{cum}$) and extinctions ($E_{cum}$) over time in observed, randomized and simulated data. We used optimal linear estimation (OLE) models[26,27] to estimate true colonization and extinction times of each species, as the raw first and last sightings are biased by the finite time frame of the time series. When OLE models estimated that colonizations probably occurred before the observation period and extinctions thereafter, the species was considered persistent. Based on all species, we calculated the net imbalance between colonizations and extinctions (NICE) over time. A perfect balance results in NICE = 0, while positive values indicate colonizations exceeding extinctions and negative values the opposite (Methods).

Across all time series, final NICE values were positive (fish: mean $NICE_{observed} \pm$ s.d. = 0.17 ± 0.8; birds: mean $NICE_{observed} \pm$ s.d. = 0.11 ± 0.7) and significantly different from zero (Student's $t_{fish}$ = 83, Student's $t_{birds}$ = 103, all $P < 0.001$) for both taxonomic groups (Fig. 4). The imbalance slightly decreased over time (LME overall slope of $NICE_{observed}$ over time for fish = $-1 \times 10^{-2}$, $P < 0.001$; and birds = $-4 \times 10^{-3}$, $P < 0.001$; Fig. 4). For simulated data, NICE values decreased over time at a slower rate than observed for birds (estimate$_{simulated}$ = $-2 \times 10^{-3}$, $P = 0.08$) while even being steady over time for fish (estimate$_{simulated}$ = $-3 \times 10^{-3}$, $P < 0.001$; Supplementary Figs. 5 and 6). Decreases in NICE values suggest that imbalances between $C_{cum}$ and $E_{cum}$ might disappear if environmental changes stop. However, the difference between observed and simulated trends in NICE suggests that extinctions are catching up with colonizations faster than predicted, which would ultimately further increase the negative deviation from the neutral prediction.

Our analyses have major implications for our understanding of biodiversity changes, but also for monitoring strategies, assessments and the formulation of conservation targets, including a reinterpretation of the 'neutral trend in richness' meta-analyses[4,10,14,15]. If most of the temporal data in these analyses have some degree of autocorrelation coupled with strong dispersal limitation, the reported zero slope does not necessarily imply constant levels of richness, but a deviation trajectory. For fish, this suggests that either colonization does not happen as fast as expected under the extinction regime, or extinction is faster than expected at the level of colonization observed. This turns the main outcome of these meta-analyses into a message of potential biodiversity decline, as the neutral prediction for changes is not necessarily a zero slope, at least for time series that are characterized by ongoing environmental change, such as climate change that changes composition by allowing colonization by 'winners' and extinction of 'losers'.

We used freshwater fish as an empirical example, as they are among the most threatened taxa[28] and are especially sensitive to their environment, but also strongly constrained by the hydrological network,

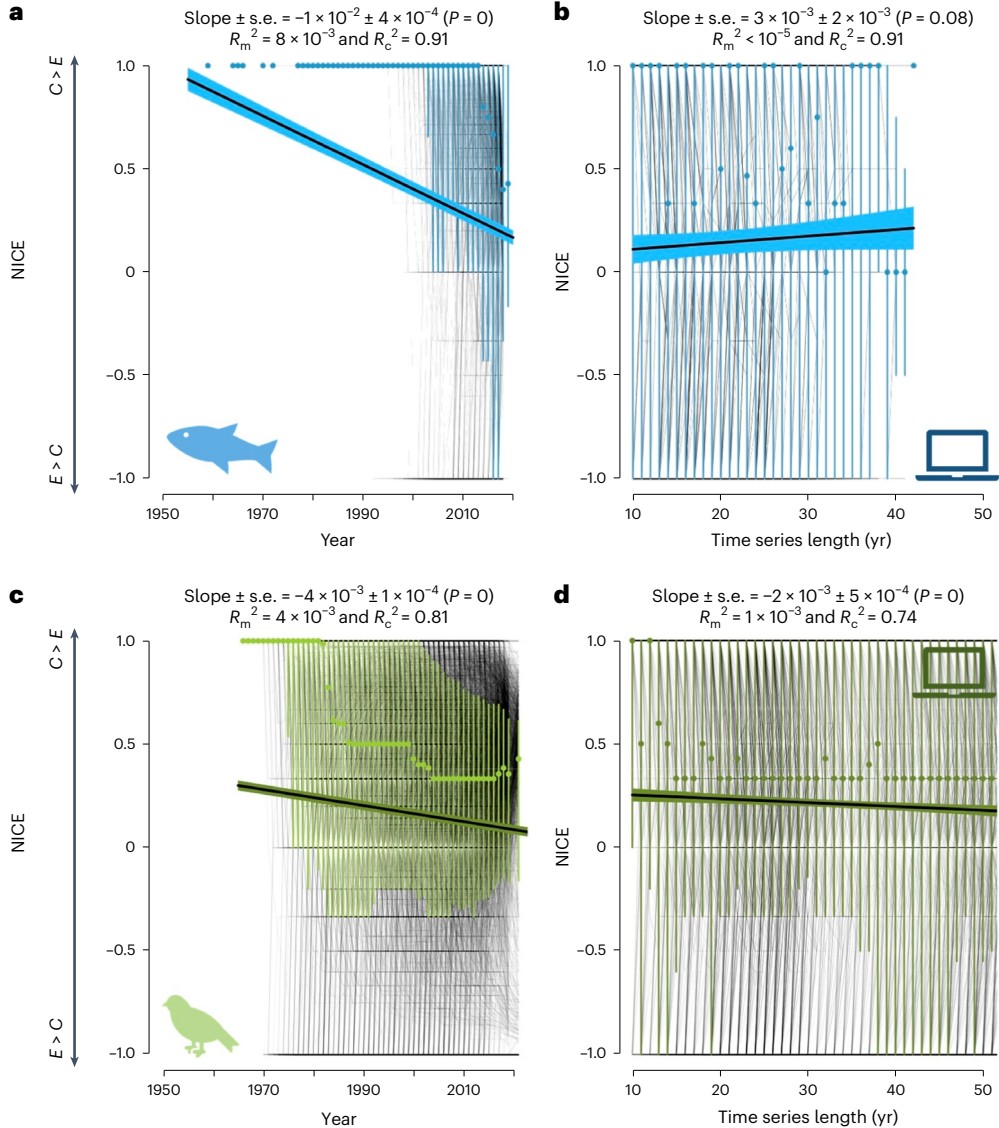

**Fig. 4 | Temporal trends in the imbalance between colonizations and extinctions measured as the NICE metric over time for observed and simulated freshwater fish and breeding birds. a**–**d**, Temporal trends in the imbalance between colonizations (*C*) and extinctions (*E*) measured as the NICE metric over time for observed (**a**,**c**) and simulated (**b**,**d**) freshwater fish (**a**,**b**) and breeding birds (**c**,**d**) time series. Background lines are the time series while the dark lines are the output of the LME models (estimate ± s.e.) from which estimates and goodness-of-fit are indicated on each panel. Points indicate the median value for each year while the associated bars represent the 25th and 75th quantiles to better represent the distributions of NICE values over time.

making escaping unsuitable conditions difficult[29]. We found simulations suggest that this increase in species richness is not fast enough to reflect long-term balanced extinction–colonization dynamics. As fish communities seem to experience sub-optimal, albeit suitable, conditions, exclusion of species is likely to take time, especially if the environment changes marginally, resulting in conditions not too far from the species optimum. The colonizers' origin was beyond the scope of this paper, but non-native species pose a critical threat to freshwater native communities[30,31] that can eventually result in increased rates of extinction[32]. Thus, considering species' origins will probably provide further insights regarding diversity dynamics and the underlying drivers[33]. On the other hand, based on our simulation for avian communities, neutral species richness trends were equal to zero, meaning that North American bird communities are experiencing an actual increase in species number. Birds being good long-distance dispersers, avian community dynamics can be strongly impacted by rescue effects. Thus, extinctions are probably evened out, although new colonizations,

for instance, by non-native species, are unlikely to fully compensate for functional loss from the native extinctions[34]. However, also based on neutral predictions, we found that extinctions are catching up with colonizations faster than expected. Thus, although for now bird communities are experiencing an increase in species richness, these temporal dynamics might be hindered by an increasing relative rate in extinctions, ultimately resulting in this increase in species number being only a transient state.

As our simulations show that richness increases by colonization–extinction imbalance are transient, they do not contradict key dynamic equilibrium theories such as the island biogeography theory[2] and the unified neutral theory of biodiversity and biogeography[1]. However, the more autocorrelated the population dynamics were, the more the imbalance between colonizations and extinctions was critical. We are not the first to report on such extended presence of non-equilibrium richness[35], but we place this idea into the context of biodiversity response to continuing and unidirectional environmental change

(for example, urbanization, climate change). The transient imbalance is likely to be shifted towards colonization and lead to richness gain. This incomplete species sorting over time will be more extensive for more long-lived organisms[36] and more dispersal constrained taxa, which are thus likely to experience the mismatch between their ecological niche and the environment for longer. However, extinctions will probably eventually catch up with colonizations when environmental conditions stop changing or when further colonization is impaired by the limited size of the species pool[37].

Delays in trends in species richness can emerge from biases and/ or actual biological processes (for example, phenotypic plasticity, use of microrefugia), resulting in imbalance between colonizations and extinctions. Although empirical data can be anywhere along the spectrum—from ecological mechanisms being the only source of bias (for example, extinction debts) to purely methodological biases—the use of neutral baselines to infer temporal trends allows potential sources to be ruled out by having ecologically null predicted trends[19]. In particular, here our neutral model allowed us to compare empirical data with null predictions to draw the following conclusions: (1) fish communities are experiencing a slower increase in diversity than expected; and (2) avian communities are exhibiting an actual increase in species richness with no apparent delays. Complementarily, NICE temporal dynamics can offer us insights regarding the ecological mechanisms underlying delays in trends, namely the imbalance between colonizations and extinctions. For instance, we showed here that although birds are not experiencing delays in species richness changes, this might be a transient pattern, given the negative trends in NICE values over time. The simultaneous use of neutral models and simple yet straightforward metrics such as NICE can allow us to disentangle mechanisms impacting species richness trend estimation.

Providing methods to quantify an accurate baseline to correct species richness trends for their inherent positive bias remains a challenge. Classically, null models remove all temporal autocorrelation in species temporal fluctuations in occurrences[21]. They are used to characterize the impact of long-term environmental changes (for example, climate change) or regular disturbance regimes (for example, tide-related disturbances, El Niño cycles) on communities and their diversity[38–40]. Although these null models provide a baseline in which environmental forcing, dispersal and species interaction effects are all simultaneously removed[21], in the context of compositional time series they delete a key constraint to our understanding of biodiversity trends: the temporal dependence of species abundances. Therefore, our simulations are neutral as species do not interact, but their dynamics are constrained by changes in population growth rates. While the null model with no temporal autocorrelation shows expected species richness trends equal to zero, the temporal constraint on population dynamics leads to a new baseline of increasing species richness, even when there is no environmental forcing. Additionally, the environmental trends are often neither white noise nor random walks, but show some aspect of autocorrelation as well[41]. The bias introduced to richness trends by the difference between colonization and extinction timing cannot be remedied with a single correction factor, as the amount of bias will differ between sites and organisms. More isolated sites will show less bias towards immigration[6], while longer-lived organisms will show more extensive extinction debt as individual generations persist longer[36]. We propose here the analysis of the NICE metric as a tool to—at least—estimate the extent of this bias, which allows comparing the contribution of trends pre-imposed by continuous environmental changes with the overall trends across empirical time series.

## Methods

### Empirical time series

To describe community dynamics over time, we used two highly curated databases. First, the RivFishTIME database, which gathers freshwater fish abundance time series[12]. We focused our analysis on 3,036 European time series with at least 10 years sampled. The final dataset comprised time series starting in 1951 and finishing in 2019 with 12 sampled years on average (s.d. = 6.6 years). Second, we used the North American Breeding Bird Survey database[13] which represents 4,317 time series sampled at least 10 times, comprising time series starting in 1966 and finishing in 2021 (29 sampled years on average ±12.5 years).

### NICE over time

As initial metrics, we estimated colonization and extinction events for each species in each time series using OLE models[26,27], using the OLE function from the sExtinct package[42], allowing for a more conservative quantification of colonization and extinction times. Although OLE models do not account for abundance dynamics, the key advantage of using them is not to rely only on the first and last sighting of a species, but rather to infer how much longer the species is likely to have persisted before and after the known occurrences. Any events (that is, colonizations and extinctions) happening outside the sampled time window of the focal community were disregarded. Thus, extinctions can theoretically happen more often than colonizations if the latter happen earlier than the beginning of the sampling time.

To compare the colonization versus extinction dynamics, we computed the NICE for each sampled year. The NICE metric quantifies the cumulative magnitude and direction of potential imbalance between local colonizations and extinctions in a comparable way across time series, and is calculated as follows:

$$\text{NICE} = \frac{C_{\text{cum}} - E_{\text{cum}}}{C_{\text{cum}} + E_{\text{cum}}}$$

Positive values indicate faster colonizations than extinctions (that is, delayed net loss), while negative values suggest slower colonizations than extinctions (that is, delayed net gain). Moreover, we estimated trends in log-transformed species richness using linear models and investigated the relationship between these trends and time series length.

### Simulated data

We used a model based on the theory of island biogeography to generate artificial data akin to the studied datasets. This model tracks the change in species richness in a site over time as follows:

$$\frac{\text{d}S_{\text{S}}}{\text{d}t} = c(S_{\text{P}} - S_{\text{S}}) - eS_{\text{S}}$$

where $S_{\text{S}}$ is the number of species in a site at a time point $t$, $S_{\text{P}}$ the number of species in the pool, and $c$ and $e$ are colonization and extinction rates, respectively. The R package island[23] implements the dynamics of this model, of which its equilibrium richness is known to be $\frac{c}{c+e}S_{\text{P}}$ and its temporal autocorrelation has been shown to be $\exp[-(c+e)\Delta t]$[22], where $\Delta t$ is the time between two consecutive samplings (which defaults to 1 for simplicity in our case). The above model is easily solved for a single species[43], leading to a Markov chain with two states for the species, which can be either present (1) or absent (0), and known transition probabilities between these states. Assuming that all species are equivalent and independent, we can obtain the temporal dynamics of a community, given its initial richness, number of species in the pool, and colonization and extinction rates. These rates have been based on the empirical data as the number of colonization events over a time series divided by the length of the time series. Thus, we simulated 9,999 time series of presence–absence data using function PA_simulation from R package island, for a species pool randomly drawn from the distribution of total number of species observed for a given time series, and time series length and initial species richness sampled at random from the observed distribution of these values in the empirical databases. As a null model, we assumed that

$c = e$, that is, the probability of any species of being present was 0.5, and a varying degree of temporal autocorrelation, which allowed us to examine the effect of transient dynamics on the model. The simulated data presented in the main text refers to an autocorrelation based on observed $c$ and $e$ in the empirical data. Moreover, we explored different imbalances between colonizations and extinctions. We focused only on the balanced rates in the main text, but results based on non-equal rates can be found in the Supplementary Information.

### Effect of time series length on species richness trends over time

To assess the potential effect of time series length on log-transformed species richness trends, we used a GAMLSS[16], which offers a highly flexible framework with regard to the response variable distribution while allowing for fitting distribution parameters as a function of the independent variable. Thus, both the mean and the variance of first the species richness trends and second the NICE values can be modelled as a linear function of time.

### Reporting summary

Further information on research design is available in the Nature Portfolio Reporting Summary linked to this article.

## Data availability

All data used in this study were attained from publicly available databases and the sources of all data and links to databases are provided at the appropriate section in the manuscript. Processed data are available on GitHub at https://github.com/Lucie-KCZ/NeutralDynamics.

## Code availability

The code is available on GitHub at https://github.com/Lucie-KCZ/NeutralDynamics.

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

## Acknowledgements

We are grateful to J. Freund, M. Pinsky, A. Ryabov and M. Rillo for their helpful comments on previous versions of the manuscript. H.H. and L.K. further acknowledge funding by the German Science Foundation funded research unit DynaCom (DFG HI 848/26-1). V.J.O. was supported by a Margarita Salas grant funded by the Spanish Ministry of Universities and the 'European Union—NextGenerationEU'.

## Author contributions

H.H. and L.K. developed the original idea and designed the research. L.K. gathered and analysed data, including the simulated data produced by V.J.O. The manuscript was written by L.K. and H.H. with contributions from V.J.O.

## Competing interests

The authors declare no competing interests.

## Additional information

**Correspondence and requests for materials** should be addressed to Lucie Kuczynski.

|---|---|

# Reporting Summary

## Statistics

For all statistical analyses, confirm that the following items are present in the figure legend, table legend, main text, or Methods section.

| n/a | Confirmed | |
|---|---|---|
| ☐ | ☒ | The exact sample size (*n*) for each experimental group/condition, given as a discrete number and unit of measurement |
| ☐ | ☒ | A statement on whether measurements were taken from distinct samples or whether the same sample was measured repeatedly |
| ☐ | ☒ | The statistical test(s) used AND whether they are one- or two-sided<br>*Only common tests should be described solely by name; describe more complex techniques in the Methods section.* |
| ☐ | ☒ | A description of all covariates tested |
| ☐ | ☒ | A description of any assumptions or corrections, such as tests of normality and adjustment for multiple comparisons |
| ☒ | ☐ | A full description of the statistical parameters including central tendency (e.g. means) or other basic estimates (e.g. regression coefficient) AND variation (e.g. standard deviation) or associated estimates of uncertainty (e.g. confidence intervals) |
| ☐ | ☒ | For null hypothesis testing, the test statistic (e.g. *F*, *t*, *r*) with confidence intervals, effect sizes, degrees of freedom and *P* value noted<br>*Give P values as exact values whenever suitable.* |
| ☒ | ☐ | For Bayesian analysis, information on the choice of priors and Markov chain Monte Carlo settings |
| ☒ | ☐ | For hierarchical and complex designs, identification of the appropriate level for tests and full reporting of outcomes |
| ☐ | ☒ | Estimates of effect sizes (e.g. Cohen's *d*, Pearson's *r*), indicating how they were calculated |

*Our web collection on statistics for biologists contains articles on many of the points above.*

## Software and code

Policy information about availability of computer code

| Data collection | NA |
|---|---|
| Data analysis | R |

For manuscripts utilizing custom algorithms or software that are central to the research but not yet described in published literature, software must be made available to editors and reviewers. We strongly encourage code deposition in a community repository (e.g. GitHub). See the Nature Portfolio guidelines for submitting code & software for further information.

## Data

Policy information about availability of data

All manuscripts must include a data availability statement. This statement should provide the following information, where applicable:

- Accession codes, unique identifiers, or web links for publicly available datasets
- A description of any restrictions on data availability
- For clinical datasets or third party data, please ensure that the statement adheres to our policy

*Provide your data availability statement here.*

# Research involving human participants, their data, or biological material

Policy information about studies with [human participants or human data](). See also policy information about [sex, gender (identity/presentation), and sexual orientation]() and [race, ethnicity and racism]().

| | |
|---|---|
| Reporting on sex and gender | Use the terms sex (biological attribute) and gender (shaped by social and cultural circumstances) carefully in order to avoid confusing both terms. Indicate if findings apply to only one sex or gender; describe whether sex and gender were considered in study design; whether sex and/or gender was determined based on self-reporting or assigned and methods used. Provide in the source data disaggregated sex and gender data, where this information has been collected, and if consent has been obtained for sharing of individual-level data; provide overall numbers in this Reporting Summary. Please state if this information has not been collected. Report sex- and gender-based analyses where performed, justify reasons for lack of sex- and gender-based analysis. |
| Reporting on race, ethnicity, or other socially relevant groupings | Please specify the socially constructed or socially relevant categorization variable(s) used in your manuscript and explain why they were used. Please note that such variables should not be used as proxies for other socially constructed/relevant variables (for example, race/ethnicity should not be used as a proxy for socioeconomic status). Provide clear definitions of the relevant terms used, how they were provided (by the participants/respondents, the researchers, or third parties), and the method(s) used to classify people into the different categories (e.g. self-report, census or administrative data, social media data, etc.) Please provide details about how you controlled for confounding variables in your analyses. |
| Population characteristics | Describe the covariate-relevant population characteristics of the human research participants (e.g. age, genotypic information, past and current diagnosis and treatment categories). If you filled out the behavioural & social sciences study design questions and have nothing to add here, write "See above." |
| Recruitment | Describe how participants were recruited. Outline any potential self-selection bias or other biases that may be present and how these are likely to impact results. |
| Ethics oversight | Identify the organization(s) that approved the study protocol. |

Note that full information on the approval of the study protocol must also be provided in the manuscript.

# Field-specific reporting

Please select the one below that is the best fit for your research. If you are not sure, read the appropriate sections before making your selection.

☐ Life sciences ☐ Behavioural & social sciences ☒ Ecological, evolutionary & environmental sciences

For a reference copy of the document with all sections, see [nature.com/documents/nr-reporting-summary-flat.pdf](http://nature.com/documents/nr-reporting-summary-flat.pdf)

# Life sciences study design

All studies must disclose on these points even when the disclosure is negative.

| | |
|---|---|
| Sample size | Describe how sample size was determined, detailing any statistical methods used to predetermine sample size OR if no sample-size calculation was performed, describe how sample sizes were chosen and provide a rationale for why these sample sizes are sufficient. |
| Data exclusions | Describe any data exclusions. If no data were excluded from the analyses, state so OR if data were excluded, describe the exclusions and the rationale behind them, indicating whether exclusion criteria were pre-established. |
| Replication | Describe the measures taken to verify the reproducibility of the experimental findings. If all attempts at replication were successful, confirm this OR if there are any findings that were not replicated or cannot be reproduced, note this and describe why. |
| Randomization | Describe how samples/organisms/participants were allocated into experimental groups. If allocation was not random, describe how covariates were controlled OR if this is not relevant to your study, explain why. |
| Blinding | Describe whether the investigators were blinded to group allocation during data collection and/or analysis. If blinding was not possible, describe why OR explain why blinding was not relevant to your study. |

# Behavioural & social sciences study design

All studies must disclose on these points even when the disclosure is negative.

| | |
|---|---|
| Study description | Briefly describe the study type including whether data are quantitative, qualitative, or mixed-methods (e.g. qualitative cross-sectional, quantitative experimental, mixed-methods case study). |
| Research sample | State the research sample (e.g. Harvard university undergraduates, villagers in rural India) and provide relevant demographic information (e.g. age, sex) and indicate whether the sample is representative. Provide a rationale for the study sample chosen. For studies involving existing datasets, please describe the dataset and source. |

| Sampling strategy | *Describe the sampling procedure (e.g. random, snowball, stratified, convenience). Describe the statistical methods that were used to predetermine sample size OR if no sample-size calculation was performed, describe how sample sizes were chosen and provide a rationale for why these sample sizes are sufficient. For qualitative data, please indicate whether data saturation was considered, and what criteria were used to decide that no further sampling was needed.* |
|---|---|
| Data collection | *Provide details about the data collection procedure, including the instruments or devices used to record the data (e.g. pen and paper, computer, eye tracker, video or audio equipment) whether anyone was present besides the participant(s) and the researcher, and whether the researcher was blind to experimental condition and/or the study hypothesis during data collection.* |
| Timing | *Indicate the start and stop dates of data collection. If there is a gap between collection periods, state the dates for each sample cohort.* |
| Data exclusions | *If no data were excluded from the analyses, state so OR if data were excluded, provide the exact number of exclusions and the rationale behind them, indicating whether exclusion criteria were pre-established.* |
| Non-participation | *State how many participants dropped out/declined participation and the reason(s) given OR provide response rate OR state that no participants dropped out/declined participation.* |
| Randomization | *If participants were not allocated into experimental groups, state so OR describe how participants were allocated to groups, and if allocation was not random, describe how covariates were controlled.* |

# Ecological, evolutionary & environmental sciences study design

All studies must disclose on these points even when the disclosure is negative.

| Study description | 4317 time series for North American breeding birds + 3036 time series for Euopean freshwater fish |
|---|---|
| Research sample | Birds data were extracted from the BBS dataset while the fish data were extracted from the RivFishTIME database |
| Sampling strategy | NA |
| Data collection | NA |
| Timing and spatial scale | Time series cover 1951-2019 (fish) and 1966-2021 (birds) and represent a local site |
| Data exclusions | NA |
| Reproducibility | NA |
| Randomization | NA |
| Blinding | NA |

Did the study involve field work? ☐ Yes ☒ No

## Field work, collection and transport

| Field conditions | *Describe the study conditions for field work, providing relevant parameters (e.g. temperature, rainfall).* |
|---|---|
| Location | *State the location of the sampling or experiment, providing relevant parameters (e.g. latitude and longitude, elevation, water depth).* |
| Access & import/export | *Describe the efforts you have made to access habitats and to collect and import/export your samples in a responsible manner and in compliance with local, national and international laws, noting any permits that were obtained (give the name of the issuing authority, the date of issue, and any identifying information).* |
| Disturbance | *Describe any disturbance caused by the study and how it was minimized.* |

# Reporting for specific materials, systems and methods

We require information from authors about some types of materials, experimental systems and methods used in many studies. Here, indicate whether each material, system or method listed is relevant to your study. If you are not sure if a list item applies to your research, read the appropriate section before selecting a response.

## Materials & experimental systems

| n/a | Involved in the study |
|-----|-----------------------|
| ☒ ☐ | Antibodies |
| ☒ ☐ | Eukaryotic cell lines |
| ☒ ☐ | Palaeontology and archaeology |
| ☒ ☐ | Animals and other organisms |
| ☒ ☐ | Clinical data |
| ☒ ☐ | Dual use research of concern |
| ☒ ☐ | Plants |

## Methods

| n/a | Involved in the study |
|-----|-----------------------|
| ☒ ☐ | ChIP-seq |
| ☒ ☐ | Flow cytometry |
| ☒ ☐ | MRI-based neuroimaging |

# Antibodies

| | |
|---|---|
| Antibodies used | *Describe all antibodies used in the study; as applicable, provide supplier name, catalog number, clone name, and lot number.* |
| Validation | *Describe the validation of each primary antibody for the species and application, noting any validation statements on the manufacturer's website, relevant citations, antibody profiles in online databases, or data provided in the manuscript.* |

# Eukaryotic cell lines

Policy information about cell lines and Sex and Gender in Research

| | |
|---|---|
| Cell line source(s) | *State the source of each cell line used and the sex of all primary cell lines and cells derived from human participants or vertebrate models.* |
| Authentication | *Describe the authentication procedures for each cell line used OR declare that none of the cell lines used were authenticated.* |
| Mycoplasma contamination | *Confirm that all cell lines tested negative for mycoplasma contamination OR describe the results of the testing for mycoplasma contamination OR declare that the cell lines were not tested for mycoplasma contamination.* |
| Commonly misidentified lines (See ICLAC register) | *Name any commonly misidentified cell lines used in the study and provide a rationale for their use.* |

# Palaeontology and Archaeology

| | |
|---|---|
| Specimen provenance | *Provide provenance information for specimens and describe permits that were obtained for the work (including the name of the issuing authority, the date of issue, and any identifying information). Permits should encompass collection and, where applicable, export.* |
| Specimen deposition | *Indicate where the specimens have been deposited to permit free access by other researchers.* |
| Dating methods | *If new dates are provided, describe how they were obtained (e.g. collection, storage, sample pretreatment and measurement), where they were obtained (i.e. lab name), the calibration program and the protocol for quality assurance OR state that no new dates are provided.* |

☐ Tick this box to confirm that the raw and calibrated dates are available in the paper or in Supplementary Information.

| | |
|---|---|
| Ethics oversight | *Identify the organization(s) that approved or provided guidance on the study protocol, OR state that no ethical approval or guidance was required and explain why not.* |

Note that full information on the approval of the study protocol must also be provided in the manuscript.

# Animals and other research organisms

Policy information about studies involving animals; ARRIVE guidelines recommended for reporting animal research, and Sex and Gender in Research

| | |
|---|---|
| Laboratory animals | *For laboratory animals, report species, strain and age OR state that the study did not involve laboratory animals.* |
| Wild animals | *Provide details on animals observed in or captured in the field; report species and age where possible. Describe how animals were caught and transported and what happened to captive animals after the study (if killed, explain why and describe method; if released, say where and when) OR state that the study did not involve wild animals.* |
| Reporting on sex | *Indicate if findings apply to only one sex; describe whether sex was considered in study design, methods used for assigning sex. Provide data disaggregated for sex where this information has been collected in the source data as appropriate; provide overall* |

*numbers in this Reporting Summary. Please state if this information has not been collected.  Report sex-based analyses where performed, justify reasons for lack of sex-based analysis.*

Field-collected samples | *For laboratory work with field-collected samples, describe all relevant parameters such as housing, maintenance, temperature, photoperiod and end-of-experiment protocol OR state that the study did not involve samples collected from the field.*

Ethics oversight | *Identify the organization(s) that approved or provided guidance on the study protocol, OR state that no ethical approval or guidance was required and explain why not.*

Note that full information on the approval of the study protocol must also be provided in the manuscript.

# Clinical data

Policy information about clinical studies

All manuscripts should comply with the ICMJE guidelines for publication of clinical research and a completed CONSORT checklist must be included with all submissions.

Clinical trial registration | *Provide the trial registration number from ClinicalTrials.gov or an equivalent agency.*

Study protocol | *Note where the full trial protocol can be accessed OR if not available, explain why.*

Data collection | *Describe the settings and locales of data collection, noting the time periods of recruitment and data collection.*

Outcomes | *Describe how you pre-defined primary and secondary outcome measures and how you assessed these measures.*

# Dual use research of concern

Policy information about dual use research of concern

## Hazards

Could the accidental, deliberate or reckless misuse of agents or technologies generated in the work, or the application of information presented in the manuscript, pose a threat to:

No | Yes

☐ ☐ Public health

☐ ☐ National security

☐ ☐ Crops and/or livestock

☐ ☐ Ecosystems

☐ ☐ Any other significant area

## Experiments of concern

Does the work involve any of these experiments of concern:

No | Yes

☐ ☐ Demonstrate how to render a vaccine ineffective

☐ ☐ Confer resistance to therapeutically useful antibiotics or antiviral agents

☐ ☐ Enhance the virulence of a pathogen or render a nonpathogen virulent

☐ ☐ Increase transmissibility of a pathogen

☐ ☐ Alter the host range of a pathogen

☐ ☐ Enable evasion of diagnostic/detection modalities

☐ ☐ Enable the weaponization of a biological agent or toxin

☐ ☐ Any other potentially harmful combination of experiments and agents

# Plants

Seed stocks | *Report on the source of all seed stocks or other plant material used. If applicable, state the seed stock centre and catalogue number. If plant specimens were collected from the field, describe the collection location, date and sampling procedures.*

Novel plant genotypes | *Describe the methods by which all novel plant genotypes were produced. This includes those generated by transgenic approaches, gene editing, chemical/radiation-based mutagenesis and hybridization. For transgenic lines, describe the transformation method, the number of independent lines analyzed and the generation upon which experiments were performed. For gene-edited lines, describe the editor used, the endogenous sequence targeted for editing, the targeting guide RNA sequence (if applicable) and how the editor*

| | was applied. |
|---|---|
| Authentication | *Describe any authentication procedures for each seed stock used or novel genotype generated. Describe any experiments used to assess the effect of a mutation and, where applicable, how potential secondary effects (e.g. second site T-DNA insertions, mosiacism, off-target gene editing) were examined.* |

# ChIP-seq

## Data deposition

☐ Confirm that both raw and final processed data have been deposited in a public database such as GEO.

☐ Confirm that you have deposited or provided access to graph files (e.g. BED files) for the called peaks.

| Data access links
*May remain private before publication.* | *For "Initial submission" or "Revised version" documents, provide reviewer access links.  For your "Final submission" document, provide a link to the deposited data.* |
|---|---|
| Files in database submission | *Provide a list of all files available in the database submission.* |
| Genome browser session
(e.g. UCSC) | *Provide a link to an anonymized genome browser session for "Initial submission" and "Revised version" documents only, to enable peer review.  Write "no longer applicable" for "Final submission" documents.* |

## Methodology

| Replicates | *Describe the experimental replicates, specifying number, type and replicate agreement.* |
|---|---|
| Sequencing depth | *Describe the sequencing depth for each experiment, providing the total number of reads, uniquely mapped reads, length of reads and whether they were paired- or single-end.* |
| Antibodies | *Describe the antibodies used for the ChIP-seq experiments; as applicable, provide supplier name, catalog number, clone name, and lot number.* |
| Peak calling parameters | *Specify the command line program and parameters used for read mapping and peak calling, including the ChIP, control and index files used.* |
| Data quality | *Describe the methods used to ensure data quality in full detail, including how many peaks are at FDR 5% and above 5-fold enrichment.* |
| Software | *Describe the software used to collect and analyze the ChIP-seq data. For custom code that has been deposited into a community repository, provide accession details.* |

# Flow Cytometry

## Plots

Confirm that:

☐ The axis labels state the marker and fluorochrome used (e.g. CD4-FITC).

☐ The axis scales are clearly visible. Include numbers along axes only for bottom left plot of group (a 'group' is an analysis of identical markers).

☐ All plots are contour plots with outliers or pseudocolor plots.

☐ A numerical value for number of cells or percentage (with statistics) is provided.

## Methodology

| Sample preparation | *Describe the sample preparation, detailing the biological source of the cells and any tissue processing steps used.* |
|---|---|
| Instrument | *Identify the instrument used for data collection, specifying make and model number.* |
| Software | *Describe the software used to collect and analyze the flow cytometry data. For custom code that has been deposited into a community repository, provide accession details.* |
| Cell population abundance | *Describe the abundance of the relevant cell populations within post-sort fractions, providing details on the purity of the samples and how it was determined.* |
| Gating strategy | *Describe the gating strategy used for all relevant experiments, specifying the preliminary FSC/SSC gates of the starting cell population, indicating where boundaries between "positive" and "negative" staining cell populations are defined.* |

☐ Tick this box to confirm that a figure exemplifying the gating strategy is provided in the Supplementary Information.

# Magnetic resonance imaging

## Experimental design

| | |
|---|---|
| Design type | *Indicate task or resting state; event-related or block design.* |
| Design specifications | *Specify the number of blocks, trials or experimental units per session and/or subject, and specify the length of each trial or block (if trials are blocked) and interval between trials.* |
| Behavioral performance measures | *State number and/or type of variables recorded (e.g. correct button press, response time) and what statistics were used to establish that the subjects were performing the task as expected (e.g. mean, range, and/or standard deviation across subjects).* |

## Acquisition

| | |
|---|---|
| Imaging type(s) | *Specify: functional, structural, diffusion, perfusion.* |
| Field strength | *Specify in Tesla* |
| Sequence & imaging parameters | *Specify the pulse sequence type (gradient echo, spin echo, etc.), imaging type (EPI, spiral, etc.), field of view, matrix size, slice thickness, orientation and TE/TR/flip angle.* |
| Area of acquisition | *State whether a whole brain scan was used OR define the area of acquisition, describing how the region was determined.* |

Diffusion MRI ☐ Used ☐ Not used

## Preprocessing

| | |
|---|---|
| Preprocessing software | *Provide detail on software version and revision number and on specific parameters (model/functions, brain extraction, segmentation, smoothing kernel size, etc.).* |
| Normalization | *If data were normalized/standardized, describe the approach(es): specify linear or non-linear and define image types used for transformation OR indicate that data were not normalized and explain rationale for lack of normalization.* |
| Normalization template | *Describe the template used for normalization/transformation, specifying subject space or group standardized space (e.g. original Talairach, MNI305, ICBM152) OR indicate that the data were not normalized.* |
| Noise and artifact removal | *Describe your procedure(s) for artifact and structured noise removal, specifying motion parameters, tissue signals and physiological signals (heart rate, respiration).* |
| Volume censoring | *Define your software and/or method and criteria for volume censoring, and state the extent of such censoring.* |

## Statistical modeling & inference

| | |
|---|---|
| Model type and settings | *Specify type (mass univariate, multivariate, RSA, predictive, etc.) and describe essential details of the model at the first and second levels (e.g. fixed, random or mixed effects; drift or auto-correlation).* |
| Effect(s) tested | *Define precise effect in terms of the task or stimulus conditions instead of psychological concepts and indicate whether ANOVA or factorial designs were used.* |

Specify type of analysis: ☐ Whole brain ☐ ROI-based ☐ Both

| | |
|---|---|
| Statistic type for inference (See Eklund et al. 2016) | *Specify voxel-wise or cluster-wise and report all relevant parameters for cluster-wise methods.* |
| Correction | *Describe the type of correction and how it is obtained for multiple comparisons (e.g. FWE, FDR, permutation or Monte Carlo).* |

## Models & analysis

| n/a | Involved in the study |
|---|---|
| ☐ | ☐ Functional and/or effective connectivity |
| ☐ | ☐ Graph analysis |
| ☐ | ☐ Multivariate modeling or predictive analysis |

| | |
|---|---|
| Functional and/or effective connectivity | *Report the measures of dependence used and the model details (e.g. Pearson correlation, partial correlation, mutual information).* |
| Graph analysis | *Report the dependent variable and connectivity measure, specifying weighted graph or binarized graph,* |

Graph analysis

*subject- or group-level, and the global and/or node summaries used (e.g. clustering coefficient, efficiency, etc.).*

Multivariate modeling and predictive analysis

*Specify independent variables, features extraction and dimension reduction, model, training and evaluation metrics.*

