## [Peer Review File · Nature Ecology & Evolution]

Peer Review Information

Journal: Nature Ecology & Evolution

Manuscript Title: Biodiversity time series are biased towards increasing species richness in changing environments

Corresponding author name(s): Lucie Kuczynski

Editorial Notes:

Reviewer Comments & Decisions:

Decision Letter, initial version:

18th November 2022

Dear Dr Kuczynski,

Your manuscript entitled "Biodiversity time series are biased towards increasing species richness in changing environments" has now been seen by 3 reviewers, whose comments are copied below. The reviewers have raised a number of concerns which we would like to see addressed in a revised manuscript before we can reach a final decision regarding publication in Nature Ecology & Evolution.

In particular, we are concerned by the points raised by both Reviewers 1 and 2 about the results potentially being artefacts of the way that the analyses dealt with lagged effects. We will need to see a convincing response to these points. We also shared Reviewer 2's view that it would be valuable to analyze the original BioTIME dataset, and we encourage you to do so, but will not require it.

We therefore invite you to revise your manuscript taking into account all reviewer and editor comments. Please highlight all changes in the manuscript text file.

We are committed to providing a fair and constructive peer-review process. Do not hesitate to email me if there are specific requests from the reviewers that you believe are technically impossible or unlikely to yield a meaningful outcome.

* If you have not done so already please begin to revise your manuscript so that it conforms to our Article format instructions at <http://www.nature.com/natecolevol/info/final-submission>. Refer also to any guidelines provided in this letter.

2Please use the link below to submit your revised manuscript and related files:

[REDACTED]

Nature Ecology & Evolution is committed to improving transparency in authorship. As part of our efforts in this direction, we are now requesting that all authors identified as 'corresponding author' on published papers create and link their Open Researcher and Contributor Identifier (ORCID) with their account on the Manuscript Tracking System (MTS), prior to acceptance. ORCID helps the scientific community achieve unambiguous attribution of all scholarly contributions. You can create and link your ORCID from the home page of the MTS by clicking on 'Modify my Springer Nature account'. For more information please visit www.springernature.com/orcid.

[REDACTED]

Reviewer expertise:

Reviewer #1: Biodiversity declines

Reviewer #2: Extinction/colonization dynamics

Reviewer #3: Aquatic biodiversity declines

Reviewers' comments:

Reviewer #1 (Remarks to the Author):

The manuscript "Biodiversity time series are biased towards increasing species richness in changing

2environments” presents a combination of simulated models and re-analysis of biodiversity time series from the RivFishTime database. The manuscript makes the argument that the positive slopes detected in the data can be explained by a “a systematic bias towards an earlier detection of colonizations than of extinctions”. This argument is supported by a series of simulations of a colonization and extinction model, which includes temporal autocorrelation. The manuscript is generally well written and is focused on an important question. The most novel component of the manuscript is the comparison between observed and simulated data, and the main inference presented in the manuscript arises from this comparison. However, the description of the model is remarkably sparse, and not enough information is presented to allow the reader to understand how the bias towards detection of colonizations arises. Presumably, it is connected to the way in which temporal autocorrelation is incorporated in the model, but there is no information as to how this is achieved. I was left wondering if it was possible that autocorrelation was delaying extinctions but not colonizations, which would explain the results presented. I was also left wondering of it would be possible to distinguish that scenario from one where there were real differences between extinction and colonization rates. More detailed information is required for the reader to be able to know whether to trust the results presented. It was also not clear to me whether species abundances were taken into consideration in the extinction and colonization model (in the main text it sounded like they might be, but in the model description it wasn't clear).

Regarding the figures, it was not obvious to me from looking at Figure 2 that the simulated results were closer to observed results than the randomised data.

I found Figure 3 very difficult to interpret, and panel a in particular, wasn't clear that it contained a lot of information

Reviewer #2 (Remarks to the Author):

Dear Authors, Dear Editor,

It is with pleasure that I reviewed your paper entitled Biodiversity time series are biased towards increasing species richness in changing environments. You will see my comments below, organized as suggested by Nature Journals.

Overall, I have one major general issue with the study. I feel like the bias toward increasing species richness is not clearly conceptualized nor fully identified. Does it result from a purely methodological bias, which would emerge from improper/delayed detection of extinction and early detection of colonization, or from biological processes that emerge from long-standing persistence of population in unsuitable habitat and fast colonization of species when habitat becomes suitable ? The bias you describe is probably in-between, but it is important to disentangle because in the second case, the richness trend observed is not really biased, as the species richness increase is due to species that are actually here. In the 1st and 3rd paragraph of the introduction, you only provide context for the biological process. The question of detection is only present in the Figure 1 finally. In the end, the phenomenon is summarized by a temporal autocorrelation in the null model, but the authors seem to consider it as a methodological/statistical phenomenon without replacing it in the context of biological

3processes.

This said, the paper provides a new and important perspective on an ongoing debate about the current biodiversity crisis. It therefore represents a worth publishing study. It is just sad to not see more in depth thinking and development of the reason behind and the implication of this bias. My second general comment is that it would have been great to not only focus on one temporal dataset (although it is great), but also to propose a similar re-analysis of the times series found in other papers such as the one using Biotime datas.

Summary of the key results

The study proposes a demonstration that the apparent increase in species richness often observed in meta- analyses of species richness trends is biased by the earlier detection of colonizations than of extinctions. By detected colonization earlier than extinction, measure richness tends to stay still or even slightly increase even under (delayed) decrease in species number at local scale. First, the authors show that simulating species richness using a neutral model (under stable richness and with temporal autocorrelation) lead to significant increases of richness over time, while no trend should be expected.

The key result of this study is that observed increase of species richness through time are inevitable when the time series are strongly autocorrelated. Even more, it appears that such bias would even hide true species richness decrease when compared to expectation from a neutral model (with similar autocorrelation).

Originality and significance

Although some conceptual aspects of the study need to be clarified (see my general comment), I think that this paper is important given the ongoing debate about biodiversity trends, and will bring significant results of broad importance.

Study design, data & methodology

The authors combine observational data (RivFishTime) and simulations for their demonstration. I don't have so much to say about the choice of time series data as I agree with the authors that this dataset represents a very good set of community time-series. I think that the demonstration would have been stronger with a re-analysis of the BioTime data, and in general by using the (freely available) datasets which have been used to demonstrate the increase of local richness.

Appropriate use of statistics and treatment of uncertainties

The statistical tools and approach used are sound. Although I am not a top specialist of the null model used in the study, it did not raise any red flag during my reading. Note also that the model used has been published in MEE.

Conclusions: robustness, validity, reliability

They conclude that temporality of species richness might be largely biased, and should consider appropriate neutral baseline for richness changes.

Suggested improvements: experiments, data for possible revision

First, I missed a more detailed discussion about the causes underlying the bias described in this study. If I understand well, what you emphasize in the paper is the fact that delayed biodiversity responses

create apparent disequilibrium in the species richness trends (i.e extinction debt). So somehow instead of showing a numerical bias you show the impact of extinction debt in biodiversity assessment.

References: appropriate credit to previous work?
Yes.

Reviewer #3 (Remarks to the Author):

Overall, this is an interesting paper that takes advantage of a large fish time series database to explore how colonisation/extinction dynamics may skew assessments of biodiversity trends based on species richness.

The story is good although the text could do with clearer discussion lines. Currently the text feels a bit scattered with some repetitions that could easily be streamlined to highlight the main takeaway messages. This would also save some space to, for example: provide a clear overview of the data used and reduce the need for an extra methods section, as well as further discussion of the limitations of the study or generalisation potential to other species, other continents for example. A few additional specific suggestions below - hope these can help.

Abstract

Some simplification, namely by reducing repetitions, would help with the flow – in the abstract and in the main text. As an example in the abstract, to avoid repeating the same idea, you could keep only the simplest expression, replacing 'This apparent increase reflected a systematic bias towards an earlier detection of colonizations than of extinctions' and the end of 'these simulated time series showed significant increases of richness over time in contrast to our equilibrium-based expectation because new species established earlier than native species declined towards extinction or detection limit' by "This apparent increase reflected the fact that new species established earlier than native species declined towards extinction or detection limit" .

The tool proposed by the paper doesn't seem to figure in the abstract -is that on purpose?

Main Text

Line numbering or at least page numbering would have been helpful for reviewing. I've used Paragraphs numbering as references (hereafter P)

P1 and P14 – decline of populations take time because of plasticity but also thanks to pockets of refuge habitat. Refuge habitat also means that some species may persist at low to undetectable levels for a long time, and if conditions get better recolonise. It was unclear to me how you have dealt with ephemeral extinctions (those which are followed by subsequent recolonisations), and also exactly how you had defined colonisation and extinction - ie how long does a species need to be absent to be considered as extinct.

P3 – The conceptual frame mentions that 'under low dispersal limitation' you'd expect that colonisation

5nature portfolio

will be fast and extinctions slow. Yet environmental changes that put adverse pressures on the fish species that you investigate are often linked to significant dispersal limitation namely through habitat fragmentation whether physical or ecological (ie through contaminants). Could this be investigated or better argued.

Given that the results P9 go into details about the length of the time series, it would help the reader to get an overview of the fish database used within the main text (P4 I guess), and not just in the extra method section. Also, the text in P9 mentions the broader BIOTIME database but I found no other mention of it elsewhere so it's unclear what it relates to.

P13 provides some interesting thoughts on the interpretation of biodiversity meta- analyses. The last sentence however is unclear to me. The Dornelas et al. 2019 paper (Dornelas, M., Gotelli, N.J., Shimadzu, H., Moyes, F., Magurran, A.E. and McGill, B.J. (2019), A balance of winners and losers in the Anthropocene. *Ecol Lett*, 22: 847-854) found evidence that colonisation and extinction are both increasing, and at similar rates – and suggested that therefore this meant that biodiversity change is accelerating. Is this what you are referring to?

P15 and elsewhere – the work you present is in the context of continuing environmental change – I am still not clear at the end of the paper whether this terminology refers to an environmental change in only one direction (e.g. continuing habitat fragmentation) or simply continuing change, which could therefore also mean times of degradation followed by restoration for example.

Also in this paragraph – the very brief reflexion on 'long-lived organisms' and the 'size of the species pool' seems interesting but seemingly unfinished, it's unclear here what effects are expected – would be nice to finish the argument.

P16 and elsewhere – I was hoping to see more discussion on short versus long term time series and limitations of the dataset used that has mainly short term time series.

Extra methods

Did you account for study differences, across all data sets, for example as a random effect.

*****END*****

Author Rebuttal to Initial commentsResponse to reviewers

Reviewer #1 (Remarks to the Author):

The manuscript “Biodiversity time series are biased towards increasing species richness in changing environments” presents a combination of simulated models and re-analysis of biodiversity time series from the RivFishTime database. The manuscript makes the argument that the positive slopes detected in the data can be explained by a “a systematic bias towards an earlier detection of colonizations than of extinctions”. This argument is supported by a series of simulations of a colonization and extinction model, which includes temporal autocorrelation. The manuscript is generally well written and is focused on an important question. The most novel component of the manuscript is the comparison between observed and simulated data, and the main inference presented in the manuscript arises from this comparison.

However, the description of the model is remarkably sparse, and not enough information is presented to allow the reader to understand how the bias towards detection of colonizations arises. Presumably, it is connected to the way in which temporal autocorrelation is incorporated in the model, but there is no information as to how this is achieved.

Response: Delays in extinctions is due to the finite nature of time series which results in species necessarily colonizing a community before going extinct. The autocorrelation component of the model increases this bias as it slows down the colonization and extinctions dynamics, although colonizations still need to happen earlier than extinctions for the latter to happen at all. We have now added details about the model in the manuscript, including a new reference. The new details explain that the basic equation in the main text can be solved for a single species, obtaining the transition probabilities of a Markov chain with two states, either present (1) or absent (0). With these transition probabilities and an initial state, we can simulate the dynamics for a single species. Assuming that species are equivalent and independent, a fair assumption for a null model, we produce the dynamics of the whole community repeating the simulation for a single species as many times as there are species in the pool. Further mathematical details are available in the references and the vignette of R package ‘island’.

L326. The above model is easily solved for a single species⁴⁷, leading to a Markov Chain with two states for the species, that can be either present (1) or absent (0), and known transition probabilities between these states. Assuming that all species are equivalent and

7independent, we can obtain the temporal dynamics of a community, given its initial richness, number of species in the pool, and colonization and extinction rates. These rates have been based on the empirical data as the number of colonization events over a time series divided by the length of the time series (See Supplementary Material). Thus, we simulated 9999 time-series of presence-absence data using function PA_simulation from R package 'island', for a species pool randomly drawn from the distribution of total number of species observed for a given time series, and time-series length and initial species richness sampled at random from the observed distribution of these values in the empirical databases. As a null model, we assumed that $c = e$, that is, the probability of any species of being present was 0.5, and a varying degree of temporal autocorrelation, that allowed us to examine the effect of transient dynamics on the model. The simulated data presented in the main text refers to an autocorrelation based on observed c and e in the empirical data. Moreover, we explored different imbalance between colonizations and extinctions. We focused only on the balanced rates in the main text, but results based on non-equal rates can be found in the Supplementary Material.

I was left wondering if it was possible that autocorrelation was delaying extinctions but not colonizations, which would explain the results presented.

Response: Using our simulations, we looked at the cumulative number of colonization events and extinction events, separately, and how they correlate with the degree of temporal autocorrelation. We have added this extra analysis in the Supplementary material, which showed no correlation between the number of events and the degree autocorrelation for birds suggesting that even a relatively weak autocorrelation can result in imbalance between colonization and extinctions. However, for fish, we found that both the final number of colonizations and of extinctions were correlated with autocorrelation.

SI4a. (about fish): Final NICE values were not correlated with the relative autocorrelation ($\rho_{\text{spearman}} = -0.016$, $p = 0.9$, $n = 124975$) while being strongly indicative of the imbalance between colonization and extinction rates ($\rho_{\text{spearman}} = 0.94$, $p < 0.001$, $n = 124975$). However, the cumulative number of colonization events at the end of simulated time series was correlation with both the relative autocorrelation ($\rho_{\text{spearman}} = -0.49$, $p = 0.01$) and the imbalance in rates ($\rho_{\text{spearman}} = 0.50$, $p = 0.01$, $n = 124975$). Finally, the cumulative number of extinction events at the end of simulated

time series was also correlated with the autocorrelation ($\rho_{\text{spearman}} = -0.54$, $p = 0.006$) but not with the imbalance ($\rho_{\text{spearman}} = -0.29$, $p = 0.2$, $n = 124975$).

SI4b. (about birds): Final NICE values were not correlated with the relative autocorrelation ($\rho_{\text{spearman}} = -0.078$, $p = 0.7$, $n = 2810824$) while being strongly indicative of the imbalance between colonization and extinction rates ($\rho_{\text{spearman}} = 0.98$, $p < 0.001$, $n = 2810824$). Similarly, the cumulative number of colonization events at the end of simulated time series was not correlated with the relative autocorrelation ($\rho_{\text{spearman}} = 0.28$, $p = 0.2$) but rather with the imbalance in rates ($\rho_{\text{spearman}} = 0.57$, $p = 0.003$, $n = 2810824$). Finally, the cumulative number of extinction events at the end of simulated time series was also not correlated with the autocorrelation ($\rho_{\text{spearman}} = 0.11$, $p = 0.6$) but with the imbalance ($\rho_{\text{spearman}} = -0.59$, $p = 0.002$, $n = 2810824$).

I was also left wondering if it would be possible to distinguish that scenario from one where there were real differences between extinction and colonization rates.

Response: We have now added results of simulations for which colonizations and extinctions are not balanced as suggested. It is indeed a good way to gain more insights on the observed patterns. Playing around with the imbalance between colonization and extinctions rates did not qualitatively change the observed patterns (i.e., the temporal autocorrelation appears as the key parameter here). All results are presented in the new Supplementary Material.

More detailed information is required for the reader to be able to know whether to trust the results presented.

Response: Overall, we have added more details in both the Method section and the Supplementary Material as we have run more simulations.

L317. We used a model based on the Theory of Island Biogeography to generate artificial data akin to the studied datasets. This model tracks the change in species richness in a

site over time as follows,

$$\frac{dS_s}{dt} = c(S_p - S_s) - eS_s$$

where S_s is the number of species in a site, S_p the number of species in the pool, and c and e are colonization and extinction rates, respectively. The R package 'island'²⁸ implements the dynamics of this model, of which its equilibrium richness is known to be $\frac{c}{c+e}S_p$ and its temporal autocorrelation has been shown to be $\exp[-(c+e)\Delta t]$ ²⁷, where Δt is the time between two consecutive samplings (which defaults to 1 for simplicity in our case). The above model is easily solved for a single species⁴⁷, leading to a Markov Chain with two states for the species, that can be either present (1) or absent (0), and known transition probabilities between these states. Assuming that all species are equivalent and independent, we can obtain the temporal dynamics of a community, given its initial richness, number of species in the pool, and colonization and extinction rates. These rates have been based on the empirical data as the number of colonization events over a time series divided by the length of the time series (See Supplementary Material). Thus, we simulated 9999 time-series of presence-absence data using function PA_simulation from R package 'island', for a species pool randomly drawn from the distribution of total number of species observed for a given time series, and time-series length and initial species richness sampled at random from the observed distribution of these values in the empirical databases. As a null model, we assumed that $c = e$, that is, the probability of any species of being present was 0.5, and a varying degree of temporal autocorrelation, that allowed us to examine the effect of transient dynamics on the model. The simulated data presented in the main text refers to an autocorrelation based on observed c and e in the empirical data. Moreover, we explored different imbalance between colonizations and extinctions. We focused only on the balanced rates in the main text, but results based on non-equal rates can be found in the Supplementary Material.

SI1a. Different temporal autocorrelation settings have been tested through different simulations. Autocorrelation is based on the colonization and extinction rates: the higher the rates are, the weaker is the temporal autocorrelation (i.e., the communities are more dynamics and less predictable). In particular, temporal autocorrelation is measured on the Markov chain (the temporal series of simulated absences and presences) for a single species, and according to Puljapurkar (1997) lag-1 temporal autocorrelation is equal to $1 - (T_{01} - T_{10})$, with T_{01} and T_{10} being the transition probabilities of the Markov chain corresponding to extinction and colonization,

10respectively. We would highlight here that temporal autocorrelation is measured on the series of presences and absences, that in turn determine the number of colonization and extinction events on that time series. So, temporal autocorrelation in these simulations refer to the maintenance of absences and presences in the temporal series, that is, the probability of not observing a change of state (1 - present, 0 - absent) in the series. It is clear that when $c = e$, and therefore $T_{01} = T_{10}$, the expected time that both absences and presences are maintained in the system is equal. However, when $c \neq e$, the expected time of presence and absence are necessarily different, but autocorrelation is still measured as a property of the whole series.

First, we ran simulations using colonisation rates estimated from the empirical data (fish: 0.41 species.year⁻¹; birds: 0.99 species.year⁻¹). Using these rates, the autocorrelation is as observed in the empirical data. By multiplying the observed rates by a constant, we could simulate stronger (i.e., constant > 1) and weaker (i.e., constant < 1) temporal autocorrelation than the observed one. It is thus important to keep in mind that variation in autocorrelation is relative here. For all the following figures, the constant is 0.001 for the panels A, F, K, P and U (i.e., very low relative autocorrelation); 0.1 for the panels B, G, L, Q and V (i.e., low autocorrelation); 1 for the panels C, H, M, R and W (i.e., observed temporal autocorrelation); 10 for the panels D, I, N, S and X (i.e., high temporal autocorrelation) and, 25 for the panels E, J, O, T and Y (i.e., very high autocorrelation). Although our results were about the relative temporal autocorrelation, they allowed us to gain insights about the absolute temporal autocorrelation as well as increased relative autocorrelation still result in increase in absolute temporal autocorrelation of the first order (i.e., correlation between x_t and x_{t-1}).

SI1b. We have run different sets of simulations testing for different (im)balances between colonization and extinction rates. The five settings are the following:

- More colonizations: $COL = EXT * \frac{1}{4}$ (coded as +2 for the correlation tests)
- Slightly more colonizations: $COL = EXT * \frac{3}{4}$ (coded as +1 for the correlation tests)
- Balance: $COL = EXT$ (coded as 0 for the correlation tests)
- Slightly more extinctions: $EXT = COL * \frac{3}{4}$ (coded as -1 for the correlation tests)
- More extinctions: $EXT = COL * \frac{1}{4}$ (coded as -2 for the correlation tests)

It was also not clear to me whether species abundances were taken into consideration in the extinction and colonization model (in the main text it sounded like they might be, but in the model description it

wasn't clear).

Response: The Optimal Linear Estimation models (OLE) we used to infer colonization and extinction timing of each species are only able to account for occurrences. In that way, our framework is only relying on presence absence data.

Response: Although the neutral model used here simulates abundances variations, our framework is actually based on occurrences for both OLE and species richness trends. A future development of this framework, beyond the scope of this specific manuscript, would be to understand how dominance patterns change over time and what are the actual baselines for these. We have clarified this aspect as follow:

L299: Although OLE models do not account for abundance dynamics, the key advantage of using them is to not rely only on the first and last sight of a species but rather infer how much longer the species is likely to have persisted before and after the known occurrences.

Regarding the figures, it was not obvious to me from looking at Figure 2 that the simulated results were closer to observed results than the randomised data.

Response: It is a mistake from our side, indeed, the bias due to time series length was highlighted by the difference between the observed and simulated patterns. However, we now have rerun the simulated data which changed a little bit the take-away of this figure (now Fig 3.), namely, we do not detect any bias anymore.

I found Figure 3 very difficult to interpret, and panel a in particular, wasn't clear that it contained a lot of information

Response: We have removed the first panel, especially now that we are also comparing datasets, hoping now that the take aways about the trends in NICE are easier to get (now Fig 4).

Reviewer #2 (Remarks to the Author):

Dear Authors, Dear Editor,

It is with pleasure that I reviewed your paper entitled Biodiversity time series are biased towards increasing species richness in changing environments. You will see my comments below, organized as suggested by Nature Journals.

Overall, I have one major general issue with the study. I feel like the bias toward increasing species richness is not clearly conceptualized nor fully identified. Does it result from a purely methodological bias, which would emerge from improper/delayed detection of extinction and early detection of colonization, or from biological processes that emerge from long-standing persistence of population in unsuitable habitat and fast colonization of species when habitat becomes suitable? The bias you describe is probably in-between, but it is important to disentangle because in the second case, the richness trend observed is not really biased, as the species richness increase is due to species that are actually here. In the 1st and 3rd paragraph of the introduction, you only provide context for the biological process. The question of detection is only present in the Figure 1 finally. In the end, the phenomenon is summarized by a temporal autocorrelation in the null model, but the authors seem to consider it as a methodological/statistical phenomenon without replacing it in the context of biological processes. This said, the paper provides a new and important perspective on an ongoing debate about the current biodiversity crisis. It therefore represents a worth publishing study. It is just sad to not see more in depth thinking and development of the reason behind and the implication of this bias.

Response: Thanks for rising this critical point. We fully agree that the two mechanisms (i.e., statistical and biological) by which this delayed response in species richness can arise need to be understood if we are to properly describe the on-going diversity crisis. We have now better discussed this aspect of our study through the manuscript.

L212: Delays in trends in species richness can emerge from biases and/or from actual biological processes (e.g., phenotypic plasticity, use of microrefugia) resulting in imbalance between colonizations and extinctions. Although empirical data can be anywhere along the spectrum from ecological mechanisms being the only source of bias

(e.g., extinction debts) to purely methodological biases, the use of neutral baselines to infer temporal trends allows to rule out potential sources by having ecologically null predicted trends²⁴. In particular, here our neutral model allowed us to compare empirical data to null predictions allowing to draw the conclusions that i) fish communities are experiencing a slower increase in diversity than expected and that on the other hand, ii) avian communities exhibit an actual increase in species richness with no apparent delays. Complementarily, NICE temporal dynamics can offer us insights regarding the ecological mechanisms underlying delays in trends, namely the imbalance between colonizations and extinctions. For instance, we showed here that although birds are not experiencing delays in species richness changes, this might be a transient pattern, given the negative trends in NICE values over time. The simultaneous use of neutral models and simple yet straightforward metrics such as NICE can allow us to disentangle mechanisms impacting species richness trend estimation.

My second general comment is that it would have been great to not only focus on one temporal dataset (although it is great), but also to propose a similar re-analysis of the times series found in other papers such as the one using Biotime datas.

Response: We have not run our analyses on the BIOTIME database as the number of time series for specific taxa is usually scarce compared to the RivFishTime database. However, we fully agree with the reviewer as having another dataset to compare our results to would be useful. For this reason, we have run our analyses on the North American Breeding Bird Survey (BBS), allowing us to test our framework on about 4300 extra time series. BBS results gave mostly complementary insights in regards to our original RivFishTime based results. That was an exciting outcome, and we expanded on throughout the manuscript.

L75: We first analysed species richness trends using ca. 3000 European empirical freshwater fish community time series from the highly curated RivFishTime dataset¹¹ (average duration = 24 years), along with ca. 4300 time series from the Breeding Bird Survey in North America¹² (average duration = 37 years) (see Extra Method Section). Across the empirically sampled communities, the average slopes from the Linear Mixed Effect model were +0.02 (SE = 0.001; $p < 0.001$; marginal $R^2 = 0.002$, conditional $R^2 = 0.85$) and +0.03 (SE = 0.0001; $p < 0.001$; marginal $R^2 = 0.007$, conditional $R^2 = 0.83$) for freshwater fish and breeding bird communities, respectively (Fig. 2A and 2D).

L88: While only the variance in species richness trends was affected by time series length for freshwater fish (estimate_{slope} ± SE = $1 \cdot 10^{-5} \pm 1 \cdot 10^{-5}$, $p = 0.3$; estimate_{variance} ± SE = $-0.04 \pm 2 \cdot 10^{-3}$, $p < 0.001$; $R^2 = 0.20$), both the mean and the variance in species richness trends were impacted for birds (estimate_{slope} ± SE = $1 \cdot 10^{-5} \pm 1 \cdot 10^{-6}$, $p < 0.001$; estimate_{variance} ± SE = $-0.03 \pm 8 \cdot 10^{-4}$, $p < 0.001$; $R^2 = 0.29$) (Fig. 3A and 3D, respectively). Thus, when dispersal is not strongly constraining communities (e.g., avian communities), short time series exhibit a duration related underestimation bias in the observed trends. While we fully acknowledge the time and money needed to collect already such data²², we need to accept that most currently used worldwide long-term datasets actually capture relatively short time series^{23,24}. Therefore, our results strongly suggest that short time series potentially underestimate diversity loss as previously claimed²⁵.

L102: For both taxa specific null models, species richness was steady over time (LME, fish: estimate ± SE = $-8 \cdot 10^{-5} \pm 3 \cdot 10^{-4}$, $p = 0.8$, marginal $R^2 < 0.001$, conditional $R^2 = 0.84$; birds: estimate ± SE = $-2 \cdot 10^{-4} \pm 1 \cdot 10^{-4}$, $p = 0.2$, marginal $R^2 < 0.001$ and conditional $R^2 = 0.82$) (Fig. 2B and 2E) while the variance was reduced under long time series (fish: estimate_{variance} ± SE = $-5 \cdot 10^{-2} \pm 5 \cdot 10^{-4}$, $p < 0.001$; $R^2 = 0.30$; birds: estimate_{variance} ± SE = $-4 \cdot 10^{-2} \pm 3 \cdot 10^{-4}$, $p < 0.001$, $R^2 = 0.48$) (Fig. 3B and 3E).

L121: Despite being a neutral model, simulated time series for river fish exhibited increased species richness over time (estimate ± SE = $4 \cdot 10^{-3} \pm 9 \cdot 10^{-4}$, $p < 0.001$, marginal $R^2 < 0.001$, conditional $R^2 = 0.85$), which suggests that these fish communities are not at equilibrium with their historical context (Fig. 2C and 2F; SI2). By contrast, simulated time series for breeding birds did not show a significant deviance from neutral trends (estimate ± SE = $-3 \cdot 10^{-4} \pm 2 \cdot 10^{-4}$, $p = 0.2$, marginal $R^2 < 0.001$, conditional $R^2 = 0.80$), which may reflect that bird communities are less constrained in their dispersal allowing stronger rescue effect²⁹. The simulated slope of richness over time was significantly independent from time series length (fish: estimate_{slope} ± SE = $-6 \cdot 10^{-6} \pm 1 \cdot 10^{-5}$, $p = 0.6$, $R^2 = 0.20$; birds: estimate_{slope} ± SE = $2 \cdot 10^{-7} \pm 4 \cdot 10^{-7}$, $p = 0.7$, $R^2 = 0.48$), only variance in species richness trends decreased with longer time series (fish: estimate_{variance} ± SE = $-5 \cdot 10^{-2} \pm 9 \cdot 10^{-4}$, $p < 0.001$; birds: estimate_{variance} ± SE = $-4 \cdot 10^{-2} \pm 5 \cdot 10^{-4}$, $p < 0.001$) (Fig. 3C and 3F). This pattern holds for most of the settings of autocorrelation and balance between colonization

and extinction we have tested (SI3). Thus, the observed departure from a zero slope for simulated data, especially in the case of riverine fish, is not linked to the empirical time series being too short⁹.

L154: Across all time series, final NICE values were positive (fish: mean $NICE_{\text{observed}} \pm SD = 0.17 \pm 0.8$; birds: mean $NICE_{\text{observed}} \pm SD = 0.11 \pm 0.7$) and significantly different from zero (Student's $t_{\text{fish}} = 83$, Student's $t_{\text{birds}} = 103$, all $p < 0.001$) for both taxonomic groups. The imbalance slightly decreased over time (LME overall slope of $NICE_{\text{observed}}$ over time for fish = $-1 \cdot 10^{-2}$, $p < 0.001$; and birds = $-4 \cdot 10^{-3}$, $p < 0.001$). For simulated data, NICE values decreased over time at a slower rate than observed for birds (estimate_{simulated} = $-2 \cdot 10^{-3}$, $p = 0.08$) while even being steady over time for fish (estimate_{simulated} = $-3 \cdot 10^{-3}$, $p < 0.001$) (SI4). Decrease in NICE values suggest that imbalance between C_{cum} and E_{cum} might disappear if environmental changes stop. However, the difference between observed and simulated trends in NICE suggests that extinctions are catching up with colonizations faster than predicted, which would ultimately further increase the negative deviation from the neutral prediction.

L187: On the other hand, based on our simulation for avian communities, neutral species richness trends were equal to zero meaning that North American bird communities are experiencing an actual increase in species number. Birds being good long-distance dispersers, avian community dynamics can be strongly impacted by rescue effects. Thus, extinctions are likely evened out, although new colonisations, for instance by non-native species unlikely fully compensate functional loss from the native extinctions³⁸. However, also based on neutral predictions, we found that extinctions are catching up with colonisations faster than expected. Thus, although for now, bird communities are experiencing increase in species richness, these temporal dynamics might be hindered by increasing relative rate in extinctions ultimately resulting in this increase in species number being only a transient state.

Summary of the key results

The study proposes a demonstration that the apparent increase in species richness often observed in meta- analyses of species richness trends is biased by the earlier detection of colonizations than of extinctions. By detected colonization earlier than extinction, measure richness tends to stay still or even

16slightly increase even under (delayed) decrease in species number at local scale. First, the authors show that simulating species richness using a neutral model (under stable richness and with temporal autocorrelation) lead to significant increases of richness over time, while no trend should be expected. The key result of this study is that observed increase of species richness through time are inevitable when the time series are strongly autocorrelated. Even more, it appears that such bias would even hide true species richness decrease when compared to expectation from a neutral model (with similar autocorrelation).

Originality and significance

Although some conceptual aspects of the study need to be clarified (see my general comment), I think that this paper is important given the ongoing debate about biodiversity trends, and will bring significant results of broad importance.

Response: Thank you for your positive assessment.

Study design, data & methodology

The authors combine observational data (RivFishTime) and simulations for their demonstration. I don't have so much to say about the choice of time series data as I agree with the authors that this dataset represents a very good set of community time-series. I think that the demonstration would have been stronger with a re-analysis of the BioTime data, and in general by using the (freely available) datasets which have been used to demonstrate the increase of local richness.

Response: As suggested, we have run our analysis on the BBS dataset - See above comment for more details.

Appropriate use of statistics and treatment of uncertainties

The statistical tools and approach used are sound. Although I am not a top specialist of the null model used in the study, it did not raise any red flag during my reading. Note also that the model used has been published in MEE.

Response: Thanks for the note about the neutral model, we actually had references already in the manuscript (Ontiveros, V. J., Capitán, J. A., Arthur, R., Casamayor, E. O. & Alonso, D. Colonization and extinction rates estimated from temporal dynamics of ecological communities: The island r package. *Methods in Ecology and Evolution* **10**, 1108-1117, doi:<https://doi.org/10.1111/2041-210X.13176> (2019). & Ontiveros, V. J., Capitán, J. A., Casamayor, E. O. & Alonso, D. The characteristic time of ecological

communities. *Ecology* **102**, e03247, doi:<https://doi.org/10.1002/ecy.3247> (2021)). But we have also added one (Alonso, D., Pinyol-Gallemí, A., Alcoverro, T., & Arhutr, R. (2015). Fish community reassembly after a coral mass mortality: higher trophic groups are subject to increased rates of extinctions. *Ecology Letters*, *18*(5), 451–461, doi:<https://doi.org/10.1111/ele.12426>)

L117: We derived these time series from a neutral model^{27,28} based on the Theory of Island Biogeography¹⁴, simulating species occurrences while controlling for equilibrium richness and temporal autocorrelation.

L325. The above model is easily solved for a single species⁴⁷, leading to a Markov Chain with two states for the species, that can be either present (1) or absent (0), and known transition probabilities between these states.

Conclusions: robustness, validity, reliability

They conclude that temporality of species richness might be largely biased and should consider appropriate neutral baseline for richness changes.

Suggested improvements: experiments, data for possible revision

First, I missed a more detailed discussion about the causes underlying the bias described in this study. If I understand well, what you emphasize in the paper is the fact that delayed biodiversity responses create apparent disequilibrium in the species richness trends (i.e extinction debt). So somehow instead of showing a numerical bias you show the impact of extinction debt in biodiversity assessment.

Response: We have now expanded on the potential mechanisms underlying the identified bias.

L212: Delays in trends in species richness can emerge from biases and/or from actual biological processes (e.g., phenotypic plasticity, use of microrefugia) resulting in imbalance between colonizations and extinctions. Although empirical data can be anywhere along the spectrum from ecological mechanisms being the only source of bias (e.g., extinction debts) to purely methodological biases, the use of neutral baselines to infer temporal trends allows to rule out potential sources by having ecologically null

nature portfolio

predicted trends²⁴. In particular, here our neutral model allowed us to compare empirical data to null predictions allowing to draw the conclusions that i) fish communities are experiencing a slower increase in diversity than expected and that on the other hand, ii) avian communities exhibit an actual increase in species richness with no apparent delays. Complementarily, NICE temporal dynamics can offer us insights regarding the ecological mechanisms underlying delays in trends, namely the imbalance between colonizations and extinctions. For instance, we showed here that although birds are not experiencing delays in species richness changes, this might be a transient pattern, given the negative trends in NICE values over time. The simultaneous use of neutral models and simple yet straightforward metrics such as NICE can allow us to disentangle mechanisms impacting species richness trend estimation.

References: appropriate credit to previous work?

Yes.Reviewer #3 (Remarks to the Author):

Overall, this is an interesting paper that takes advantage of a large fish time series database to explore how colonisation/extinction dynamics may skew assessments of biodiversity trends based on species richness.

The story is good although the text could do with clearer discussion lines.

Currently the text feels a bit scattered with some repetitions that could easily be streamlined to highlight the main takeaway messages. This would also save some space to, for example: provide a clear overview of the data used and reduce the need for an extra methods section, as well as further discussion of the limitations of the study or generalisation potential to other species, other continents for example.

A few additional specific suggestions below - hope these can help.

Response: Thanks for the writing suggestions, we tried to improve text flow as much as possible following your recommendations. Moreover, as suggested by another reviewer, we actually ran our analysis on an additional dataset allowing us to discuss better the potential limitations.

Abstract

Some simplification, namely by reducing repetitions, would help with the flow – in the abstract and in the main text. As an example in the abstract, to avoid repeating the same idea, you could keep only the simplest expression, replacing ‘This apparent increase reflected a systematic bias towards an earlier detection of colonizations than of extinctions’ and the end of ‘ these simulated time series showed significant increases of richness over time in contrast to our equilibrium-based expectation because new species established earlier than native species declined towards extinction or detection limit’ by “This apparent increase reflected the fact that new species established earlier than native species declined towards extinction or detection limit” .

Response: We thoroughly went through the manuscript and reduced potential repetitions. At the suggested phrase, however, the proposed new phrasing doesn’t reduce the number of words so we opted for a different solution.

The tool proposed by the paper doesn't seem to figure in the abstract - is that on purpose?

Response: We wanted to focus on the ecological take-away and think that the neutral model is the key methodological aspect here.

Main Text

Line numbering or at least page numbering would have been helpful for reviewing. I've used Paragraphs numbering as references (hereafter P)

Response: We are really sorry about this and apologize for the inconvenience – we have added line numbers.

P1 and P14 – decline of populations take time because of plasticity but also thanks to pockets of refuge habitat. Refuge habitat also means that some species may persist at low to undetectable levels for a long time, and if conditions get better recolonise. It was unclear to me how you have dealt with ephemeral extinctions (those which are followed by subsequent recolonisations), and also exactly how you had defined colonisation and extinction - ie how long does a species need to be absent to be considered as extinct.

Response: That is a really good point and studies have defined colonization and extinction in different ways, leading to incongruent patterns. Here, we focused on local colonizations and extinctions as being the first and last colonization and extinction, respectively. Especially, as we were interested in the global species richness trends, species strongly fluctuating from presence – absence would not impact that much this global trend. Moreover, these fluctuations could only result from species detectability, in particular for rare species. One could argue that even the first and last event are likely to be influenced by species detectability, and we would fully agree with this point. That is the reasoning that got us using the OLE models, allowing to model for each species at each location (i.e., populations) its colonization and extinction timing. We clarify this point in the manuscript as it was not clear.

L57: However, under low dispersal limitation, one can assume that colonizations (**defined as the first colonization event over a given time series**) will be fast (as it needs only few propagules), whereas extinctions (**defined as the last extinction event over a given time series**) will be delayed because in the absence of catastrophic mortality population growth will slowly turn negative for the losers.

L45: As the exponential decline of existing populations takes time (e.g., because of plasticity, use of microrefugia), extinction debts will lead to a delayed reduction in richness^{16,17} and the negative richness trends will only emerge later.

P3 – The conceptual frame mentions that ‘under low dispersal limitation’ you’d expect that colonisation will be fast and extinctions slow. Yet environmental changes that put adverse pressures on the fish species that you investigate are often linked to significant dispersal limitation namely through habitat fragmentation whether physical or ecological (ie through contaminants). Could this be investigated or better argued.

Response: We fully agree that fish communities are strongly constrained in their dispersal, and thus ability to escape unsuitable environments. By investigating patterns in species richness delays on birds, we somehow touched upon this question as birds are not as dispersal limited as fish both due to their physiology (fish are strictly aquatic and ectotherms) and the environment they live in (dams are less likely to be cross by fish than a road, although being an actual threat, for birds for instance). We highlighted this difference in dispersal limitation throughout the manuscript now.

L124: By contrast, simulated time series for breeding birds did not show a significant deviance from neutral trends (estimate \$\pm\$ SE = \$-3 \cdot 10^{-4} \pm 2 \cdot 10^{-4}\$, \$p = 0.2\$, marginal \$R^2 < 0.001\$, conditional \$R^2 = 0.80\$ ), which may reflect that bird communities are less constrained in their dispersal allowing stronger rescue effect²⁹.

L167: If most of the temporal data in these analyses have some degree of autocorrelation coupled with strong dispersal limitation, the reported zero slope does not necessarily imply constant levels of richness, but a deviation trajectory.

L187: On the other hand, based on our simulation for avian communities, neutral species richness trends were equal to zero meaning that North American bird communities are experiencing an actual increase in species number. Birds being good long-distance dispersers, avian community dynamics can be strongly impacted by rescue effects. Thus,

extinctions are likely evened out, although new colonisations, for instance by non-native species unlikely fully compensate functional loss from the native extinctions³⁸.

Given that the results P9 go into details about the length of the time series, it would help the reader to get an overview of the fish database used within the main text (P4 I guess), and not just in the extra method section. Also, the text in P9 mentions the broader BIOTIME database but I found no other mention of it elsewhere so it's unclear what it relates to.

Response: We have now added the average time series duration to give a better overview of the datasets, we could not expand more for space's sake. Regarding the BIOTIME database, we initially aimed at using it as an example of a long-term database that would still suffer potential bias in species richness trend estimates based on the predictions from our neutral model. However, as it was confusing and that we have now two datasets to describe, we remove this reference.

L75: We first analysed species richness trends using ca. 3000 European empirical freshwater fish community time series from the highly curated RivFishTime dataset¹¹ (average duration = 24 years), along with ca. 4300 time series from the Breeding Bird Survey in North America¹² (average duration = 37 years) (see Extra Method Section).

P13 provides some interesting thoughts on the interpretation of biodiversity meta- analyses. The last sentence however is unclear to me. The Dornelas et al. 2019 paper (Dornelas, M., Gotelli, N.J., Shimadzu, H., Moyes, F., Magurran, A.E. and McGill, B.J. (2019), A balance of winners and losers in the Anthropocene. *Ecol Lett*, 22: 847-854) found evidence that colonisation and extinction are both increasing, and at similar rates – and suggested that therefore this meant that biodiversity change is accelerating. Is this what you are referring to?

Response: We actually do not refer to this study. However, our point is that the neutral predictions is not necessarily a zero slope for species richness trends, we clarified this as follow:

L172: This turns the main outcome of these meta-analyses into a message of potential biodiversity decline, as the neutral prediction for changes is not necessarily a zero slope, at least for time series that are characterized by ongoing environmental change such as

climate change that changes composition by allowing colonization by “winners” and extinction of “losers”.

P15 and elsewhere – the work you present is in the context of continuing environmental change – I am still not clear at the end of the paper whether this terminology refers to an environmental change in only one direction (e.g. continuing habitat fragmentation) or simply continuing change, which could therefore also mean times of degradation followed by restoration for example.

Response: Here, we mean a consistent direction for a given environmental change (e.g., continuing habitat fragmentation). We clarified this.

L160: This turns the main outcome of these meta-analyses into a message of potential biodiversity decline, as the neutral prediction for changes is not necessarily a zero slope, at least for time series that are characterized by ongoing environmental change such as climate change that changes composition by allowing colonization by “winners” and extinction of “losers”.

L202: We are not the first to report on such extended presence of non-equilibrium richness³⁹, but we place this idea into the context of biodiversity response to continuing environmental change (e.g., urbanization, climate change).

L230: They are used to characterise the impact of long-term environmental changes (e.g., climate change) or regular disturbance regimes (e.g., tide-related disturbances, El Niño cycles) on communities and their diversity^{42–44}.

Also in this paragraph – the very brief reflexion on ‘long-lived organisms’ and the ‘size of the species pool’ seems interesting but seemingly unfinished, it’s unclear here what effects are expected – would be nice to finish the argument.

Response: We have expanded on these ideas as suggested.

L206: This incomplete species sorting over time will be more extensive for more long-lived organisms⁴⁰ and more dispersal constrained taxa which are thus likely to experience for longer the mismatch between their ecological niche and the environment. However, extinctions will likely eventually catch up with colonizations when environmental conditions stop changing or when further colonization is impaired by the limited size of the species pool⁴¹.

P16 and elsewhere – I was hoping to see more discussion on short versus long term time series and limitations of the dataset used that has mainly short term time series.

Response: The new simulations based on our neutral model with smaller species pool to better match our data alleviated the bias due to time series length. This results from the fact that the number of colonizations events cannot be higher than the number of species in the species pool. By reducing the species pool to a more local definition (rather than all species ever sampled across all locations), we reduce the potential number of colonizations happening before extinctions and in turn the delay in species richness trends. We have explained how the new neutral simulations were obtained in the Extra Method Section as well as in the Supplementary Material.

L317. We used a model based on the Theory of Island Biogeography to generate artificial data akin to the studied datasets. This model tracks the change in species richness in a site over time as follows,

$$\frac{dS_s}{dt} = c(S_p - S_s) - eS_s$$

where S_s is the number of species in a site, S_p the number of species in the pool, and c and e are colonization and extinction rates, respectively. The R package 'island'²⁸ implements the dynamics of this model, of which its equilibrium richness is known to be $\frac{c}{c+e} S_p$ and its temporal autocorrelation has been shown to be $\exp[-(c+e) \Delta t]$ ²⁷, where Δt is the time between two consecutive samplings (which defaults to 1 for simplicity in our case). The above model is easily solved for a single species⁴⁷, leading to a Markov Chain with two states for the species, that can be either present (1) or absent (0), and known transition probabilities between these states. Assuming that all species are equivalent and independent, we can obtain the temporal dynamics of a community, given its initial richness, number of species in the pool, and colonization and extinction rates. These rates have been based on the empirical data as the number of colonization events over a time series divided by the length of the time series (See Supplementary Material). Thus, we

simulated 9999 time-series of presence-absence data using function PA_simulation from R package 'island', for a species pool randomly drawn from the distribution of total number of species observed for a given time series, and time-series length and initial species richness sampled at random from the observed distribution of these values in the empirical databases. As a null model, we assumed that $c = e$, that is, the probability of any species of being present was 0.5, and a varying degree of temporal autocorrelation, that allowed us to examine the effect of transient dynamics on the model. The simulated data presented in the main text refers to an autocorrelation based on observed c and e in the empirical data. Moreover, we explored different imbalance between colonizations and extinctions. We focused only on the balanced rates in the main text, but results based on non-equal rates can be found in the Supplementary Material.

Supplementary material. **Description of all settings with which simulations have been run**

Different autocorrelation settings

Different temporal autocorrelation settings have been tested through different simulations. Autocorrelation is based on the colonization and extinction rates: the higher the rates are, the weaker is the temporal autocorrelation (i.e., the communities are more dynamics and less predictable). In particular, temporal autocorrelation is measured on the Markov chain (the temporal series of simulated absences and presences) for a single species, and according to Puljapurkar (1997) lag-1 temporal autocorrelation is equal to $1 - (T_{01} - T_{10})$, with T_{01} and T_{10} being the transition probabilities of the Markov chain corresponding to extinction and colonization, respectively. We would highlight here that temporal autocorrelation is measured on the series of presences and absences, that in turn determine the number of colonization and extinction events on that time series. So, temporal autocorrelation in these simulations refer to the maintenance of absences and presences in the temporal series, that is, the probability of not observing a change of state (1 - present, 0 - absent) in the series. It is clear that when $c = e$, and therefore $T_{01} = T_{10}$, the expected time that both absences and presences are maintained in the system is equal. However, when $c \neq e$, the expected time of presence and absence are necessarily different, but autocorrelation is still measured as a property of the whole series.

First, we ran simulations using colonisation rates estimated from the empirical data (fish: 0.41 species.year⁻¹; birds: 0.99 species.year⁻¹). Using these rates, the autocorrelation is as observed in the empirical data. By multiplying the observed rates by a constant, we could simulate stronger (i.e., constant > 1) and weaker (i.e., constant < 1) temporal autocorrelation than the observed

one. It is thus important to keep in mind that variation in autocorrelation is relative here. For all the following figures, the constant is 0.001 for the panels A, F, K, P and U (i.e., very low relative autocorrelation); 0.1 for the panels B, G, L, Q and V (i.e., low autocorrelation); 1 for the panels C, H, M, R and W (i.e., observed temporal autocorrelation); 10 for the panels D, I, N, S and X (i.e., high temporal autocorrelation) and, 25 for the panels E, J, O, T and Y (i.e., very high autocorrelation). Although our results were about the relative temporal autocorrelation, they allowed us to gain insights about the absolute temporal autocorrelation as well as increased relative autocorrelation still result in increase in absolute temporal autocorrelation of the first order (i.e., correlation between x_t and x_{t-1}).

Table S1.a: Range of the Spearman's correlations between SR at time t and time $t-1$ (i.e., temporal autocorrelation of the first order) across the different relative autocorrelation settings.

Relative autocorrelation	Fish		Birds	
	Minimum	Maximum	Minimum	Maximum
Very low	-0.14	-0.14	-0.06	-0.05
Low	-0.15	-0.14	-0.06	-0.05
As observed	0.06	0.23	0.02	0.14
High	0.52	0.66	0.63	0.87
Very high	0.58	0.69	0.76	0.92

Reference: Tuljapurkar S. (1997). Stochastic Matrix Models. In: *Structured Population Models in Marine, Terrestrial, and Freshwater Systems*, pages 59-87. Springer US, Boston, MA.

Different imbalance between colonisation and extinction rates

We have run different sets of simulations testing for different (im)balances between colonization and extinction rates. The five settings are the following:

- More colonizations: $COL = EXT * \frac{1}{4}$ (coded as +2 for the correlation tests)
- Slightly more colonizations: $COL = EXT * \frac{3}{4}$ (coded as +1 for the correlation tests)
- Balance: $COL = EXT$ (coded as 0 for the correlation tests)
- Slightly more extinctions: $EXT = COL * \frac{3}{4}$ (coded as -1 for the correlation tests)
- More extinctions: $EXT = COL * \frac{1}{4}$ (coded as -2 for the correlation tests)

Did you account for study differences, across all data sets, for example as a random effect.

Response: We did account for a potential study effect as we put the study ID as a random effect in our global mixed effect models. We could have only had the sampling methods, but studies might differ in more ways from only the sampling aspect and we thus included the ID instead which virtually incorporates all inter-study differences.

Decision Letter, first revision:

1st March 2023

Dear Dr. Kuczynski,

Thank you for submitting your revised manuscript "Biodiversity time series are biased towards increasing species richness in changing environments" (NATECOLEVOL-220917616A). It has now been seen again by the original reviewers and their comments are below. The reviewers find that the paper has improved in revision, and therefore we'll be happy in principle to publish it in Nature Ecology & Evolution, pending minor revisions to satisfy the reviewers' final requests and to comply with our editorial and formatting guidelines.

[REDACTED]

Reviewer #2 (Remarks to the Author):

Dear authors,

I honestly don't have so much more to say. You have answered my concerns, and you really did a good job in adding BBS trends to the analysis. And it seems that other's reviewers remarks have improved the paper.

It is really an interesting study that brings a new piece in the debate about biodiversity trends.

Reviewer #3 (Remarks to the Author):

Dear Authors,

This version I feel is much improved.

Now that the slight mismatch created by the mention of a dataset that was not utilised has been removed, and another (more relevant) dataset has been analysed, the narrative feels more logical and better evidenced. I agree that the bird results provide complementary insights compared to the fish results, and this has added value to the paper.

Regarding my own comments I feel that the text flow has improved, and that my main concerns have been addressed. Clarifications on the definition of the terms of colonization and extinction used really help. I particularly like that the additional bird data allow to investigate a bit more the role of dispersal limitation on colonisation/extinction mechanisms, and I think this is an important outcome. The new simulations based on the neutral model seem to address my request for more development on the limitations of short term time series.

Very minor suggestions:

-Regarding my comment on the directional nature of the continuing environmental change under investigation – and given that you mean 'a consistent direction for a given change', it would be good to say so rather than only rely on exemplars – so just adding referring to a 'continuing/ongoing and unidirectional environmental change.

-A minor wording suggestion on new text Line 187 - Thus, extinctions are likely to be evened out, although new colonisations, for instance by non-native species, are unlikely to fully compensate functional loss...

Our ref: NATECOLEVOL-220917616A

28th March 2023

Dear Dr. Kuczynski,

Thank you for your patience as we've prepared the guidelines for final submission of your Nature Ecology & Evolution manuscript, "Biodiversity time series are biased towards increasing species richness in changing environments" (NATECOLEVOL-220917616A). Please carefully follow the step-by-step instructions provided in the attached file, and add a response in each row of the table to indicate the changes that you have made. Please also check and comment on any additional marked-up edits we have proposed within the text. Ensuring that each point is addressed will help to ensure that your revised manuscript can be swiftly handed over to our production team.

****We would like to start working on your revised paper, with all of the requested files and forms, as soon as possible (preferably within two weeks). Please get in contact with us immediately if you anticipate it taking more than two weeks to submit these revised files.****

In recognition of the time and expertise our reviewers provide to Nature Ecology & Evolution's editorial process, we would like to formally acknowledge their contribution to the external peer review of your manuscript entitled "Biodiversity time series are biased towards increasing species richness in changing environments". For those reviewers who give their assent, we will be publishing their names alongside the published article.

Nature Ecology & Evolution offers a Transparent Peer Review option for new original research manuscripts submitted after December 1st, 2019. As part of this initiative, we encourage our authors to support increased transparency into the peer review process by agreeing to have the reviewer comments, author rebuttal letters, and editorial decision letters published as a Supplementary item. When you submit your final files please clearly state in your cover letter whether or not you would like to participate in this initiative. Please note that failure to state your preference will result in delays in accepting your manuscript for publication.

Cover suggestions

As you prepare your final files we encourage you to consider whether you have any images or

30illustrations that may be appropriate for use on the cover of Nature Ecology & Evolution.

Nature Ecology & Evolution has now transitioned to a unified Rights Collection system which will allow our Author Services team to quickly and easily collect the rights and permissions required to publish your work. Approximately 10 days after your paper is formally accepted, you will receive an email in providing you with a link to complete the grant of rights. If your paper is eligible for Open Access, our Author Services team will also be in touch regarding any additional information that may be required to arrange payment for your article.

Please note that *Nature Ecology & Evolution* is a Transformative Journal (TJ). Authors may publish their research with us through the traditional subscription access route or make their paper immediately open access through payment of an article-processing charge (APC). Authors will not be required to make a final decision about access to their article until it has been accepted. [Find out more about Transformative Journals](https://www.springernature.com/gp/open-research/transformative-journals)

Authors may need to take specific actions to achieve [compliance with funder and institutional open access mandates](https://www.springernature.com/gp/open-research/funding/policy-compliance-faqs). If your research is supported by a funder that requires immediate open access (e.g. according to [Plan S principles](https://www.springernature.com/gp/open-research/plan-s-compliance)) then you should select the gold OA route, and we will direct you to the compliant route where possible. For authors selecting the subscription publication route, the journal's standard licensing terms will need to be accepted, including [self-archiving and license to publish](https://www.nature.com/nature-portfolio/editorial-policies/self-archiving-and-license-to-publish). Those licensing terms will supersede any other terms that the author or any third party may assert apply to any version of the manuscript.

For information regarding our different publishing models please see our page

[href="https://www.springernature.com/gp/open-research/transformative-journals">](https://www.springernature.com/gp/open-research/transformative-journals) Transformative Journals page. If you have any questions about costs, Open Access requirements, or our legal forms, please contact ASJournals@springernature.com.

[REDACTED]

[REDACTED]

Reviewer #2:

Remarks to the Author:

Dear authors,

I honestly don't have so much more to say. You have answered my concerns, and you really did a good job in adding BBS trends to the analysis. And it seems that other's reviewers remarks have improved the paper.

It is really an interesting study that brings a new piece in the debate about biodiversity trends.

Reviewer #3:

Remarks to the Author:

Dear Authors,

This version I feel is much improved.

Now that the slight mismatch created by the mention of a dataset that was not utilised has been removed, and another (more relevant) dataset has been analysed, the narrative feels more logical and better evidenced. I agree that the bird results provide complementary insights compared to the fish results, and this has added value to the paper.

Regarding my own comments I feel that the text flow has improved, and that my main concerns have been addressed. Clarifications on the definition of the terms of colonization and extinction used really help. I particularly like that the additional bird data allow to investigate a bit more the role of dispersal limitation on colonisation/extinction mechanisms, and I think this is an important outcome. The new simulations based on the neutral model seem to address my request for more development on the limitations of short term time series.

Very minor suggestions:

-Regarding my comment on the directional nature of the continuing environmental change under investigation – and given that you mean 'a consistent direction for a given change', it would be good

32to say so rather than only rely on exemplars – so just adding referring to a 'continuing/ongoing and unidirectional environmental change.

-A minor wording suggestion on new text Line 187 - Thus, extinctions are likely to be evened out, although new colonisations, for instance by non-native species, are unlikely to fully compensate functional loss...

Author Rebuttal, first revision:

Dear Editor and reviewers,

We would like to sincerely thank you for your comments which really helped to improve the manuscript while being really supportive. We have now edited the manuscript with the last comments.

Reviewer #2 (Remarks to the Author):

Dear authors,

I honestly don't have so much more to say. You have answered my concerns, and you really did a good job in adding BBS trends to the analysis. And it seems that other's reviewers remarks have improved the paper. It is really an interesting study that brings a new piece in the debate about biodiversity trends.

Response: Thank you for the support through your constructive comments.

Reviewer #3 (Remarks to the Author):

Dear Authors,

This version I feel is much improved.

Now that the slight mismatch created by the mention of a dataset that was not utilised has been removed, and another (more relevant) dataset has been analysed, the narrative feels more logical and better evidenced. I agree that the bird results provide complementary insights compared to the fish results, and this has added value to the paper.

Regarding my own comments I feel that the text flow has improved, and that my main concerns have been addressed. Clarifications on the definition of the terms of colonization and extinction used really help. I particularly like that the additional bird data allow to investigate a bit more the role of dispersal limitation on colonisation/extinction mechanisms, and I think this is an important outcome. The new simulations based on the neutral model seem to address my request for more development on the limitations of short term time series.

33Response: Thank you for the support through your helpful comments.

Very minor suggestions:

-Regarding my comment on the directional nature of the continuing environmental change under investigation – and given that you mean ‘a consistent direction for a given change’, it would be good to say so rather than only rely on exemplars – so just adding referring to a ‘continuing/ongoing and unidirectional environmental change.

Response: We have edited the sentence as follow:

L210: We are not the first to report on such extended presence of non-equilibrium richness³⁵, but we place this idea into the context of biodiversity response to continuing and unidirectional environmental change (e.g., urbanization, climate change).

-A minor wording suggestion on new text Line 187 - Thus, extinctions are likely to be evened out, although new colonisations, for instance by non-native species, are unlikely to fully compensate functional loss...

Response: Done

Final Decision Letter:

We are pleased to inform you that your Article entitled "Biodiversity time series are biased towards increasing species richness in changing environments", has now been accepted for publication in Nature Ecology & Evolution.

Over the next few weeks, your paper will be copyedited to ensure that it conforms to Nature Ecology and Evolution style. Once your paper is typeset, you will receive an email with a link to choose the appropriate publishing options for your paper and our Author Services team will be in touch regarding any additional information that may be required

You will not receive your proofs until the publishing agreement has been received through our system

Due to the importance of these deadlines, we ask you please us know now whether you will be difficult to contact over the next month. If this is the case, we ask you provide us with the contact information (email, phone and fax) of someone who will be able to check the proofs on your behalf, and who will be available to address any last-minute problems . Once your paper has been scheduled for online publication, the Nature press office will be in touch to confirm the details.

Acceptance of your manuscript is conditional on all authors' agreement with our publication policies

34(see www.nature.com/authors/policies/index.html). In particular your manuscript must not be published elsewhere and there must be no announcement of the work to any media outlet until the publication date (the day on which it is uploaded onto our web site).

Please note that *Nature Ecology & Evolution* is a Transformative Journal (TJ). Authors may publish their research with us through the traditional subscription access route or make their paper immediately open access through payment of an article-processing charge (APC). Authors will not be required to make a final decision about access to their article until it has been accepted. [Find out more about Transformative Journals](https://www.springernature.com/gp/open-research/transformative-journals)

Authors may need to take specific actions to achieve [compliance with funder and institutional open access mandates](https://www.springernature.com/gp/open-research/funding/policy-compliance-faqs). If your research is supported by a funder that requires immediate open access (e.g. according to [Plan S principles](https://www.springernature.com/gp/open-research/plan-s-compliance)) then you should select the gold OA route, and we will direct you to the compliant route where possible. For authors selecting the subscription publication route, the journal's standard licensing terms will need to be accepted, including [self-archiving and license to publish](https://www.nature.com/nature-portfolio/editorial-policies/self-archiving-and-license-to-publish). Those licensing terms will supersede any other terms that the author or any third party may assert apply to any version of the manuscript.

We welcome the submission of potential cover material (including a short caption of around 40 words) related to your manuscript; suggestions should be sent to Nature Ecology & Evolution as electronic files (the image should be 300 dpi at 210 x 297 mm in either TIFF or JPEG format). Please note that such pictures should be selected more for their aesthetic appeal than for their scientific content, and that colour images work better than black and white or grayscale images. Please do not try to design a cover with the Nature Ecology & Evolution logo etc., and please do not submit composites of images related to your work. I am sure you will understand that we cannot make any promise as to whether any of your suggestions might be selected for the cover of the journal.

nature portfolio

You can generate the link yourself when you receive your article DOI by entering it here: <http://authors.springernature.com/share>.

Yours sincerely,

Alexa

Alexa McKay, PhD
Senior Editor
Nature Ecology and Evolution

P.S. Click on the following link if you would like to recommend Nature Ecology & Evolution to your librarian <http://www.nature.com/subscriptions/recommend.html#forms>

** Visit the Springer Nature Editorial and Publishing website at http://editorial-jobs.springernature.com?utm_source=ejP_NEcoE_email&utm_medium=ejP_NEcoE_email&utm_campaign=ejP_NEcoE for more information about our career opportunities. If you have any questions please click [here](mailto:editorial.publishing.jobs@springernature.com).**